# Mechanistic analysis of Riboswitch Ligand interactions provides insights into pharmacological control over gene expression

Shaifaly Parmar[1], Desta Doro Bume[1], Colleen M. Connelly [1], Robert E. Boer[1], Peri R. Prestwood[1], Zhen Wang[2], Henning Labuhn[2], Krishshanthi Sinnadurai[2], Adeline Feri[2], Jimmy Ouellet[2], Philip Homan[3,4], Tomoyuki Numata[5] & John S. Schneekloth Jr. [1] ✉

Riboswitches are structured RNA elements that regulate gene expression upon binding to small molecule ligands. Understanding the mechanisms by which small molecules impact riboswitch activity is key to developing potent, selective ligands for these and other RNA targets. We report the structure-informed design of chemically diverse synthetic ligands for $PreQ_1$ riboswitches. Multiple X-ray co-crystal structures of synthetic ligands with the *Thermoanaerobacter tengcongensis* (*Tte*)-$PreQ_1$ riboswitch confirm a common binding site with the cognate ligand, despite considerable chemical differences among the ligands. Structure probing assays demonstrate that one ligand causes conformational changes similar to $PreQ_1$ in six structurally and mechanistically diverse $PreQ_1$ riboswitch aptamers. Single-molecule force spectroscopy is used to demonstrate differential modes of riboswitch stabilization by the ligands. Binding of the natural ligand brings about the formation of a persistent, folded pseudoknot structure, whereas a synthetic ligand decreases the rate of unfolding through a kinetic mechanism. Single round transcription termination assays show the biochemical activity of the ligands, while a GFP reporter system reveals compound activity in regulating gene expression in live cells without toxicity. Taken together, this study reveals that diverse small molecules can impact gene expression in live cells by altering conformational changes in RNA structures through distinct mechanisms.

RNA molecules exert function through specific sequences capable of adopting diverse secondary and tertiary structures[1,2]. The formation of dynamic structural ensembles by RNA is governed by a range of factors, including inherent thermodynamic properties as well as contexts and cues within the cellular environment such as protein or small molecule binding[3]. Within biological contexts, it is frequently the case that an RNA will adopt an ensemble of different but energetically similar three-dimensional structures that coexist simultaneously[4,5].

[1]Chemical Biology Laboratory, Center for Cancer Research, National Cancer Institute, Frederick, MD, USA. [2]Depixus SAS, Paris, France. [3]Center for Cancer Research Collaborative Bioinformatics Resource, National Cancer Institute, National Institutes of Health, Bethesda, MD, USA. [4]Advanced Biomedical Computational Science, Frederick National Laboratory for Cancer Research, Frederick, MD, USA. [5]Department of Bioscience and Biotechnology, Graduate School of Bioresource and Bioenvironmental Sciences, Kyushu University, Fukuoka, Japan. ✉e-mail: schneeklothjs@mail.nih.gov

Despite this conformational heterogeneity, many RNAs are still capable of folding into structures with hydrophobic pockets that are likely targetable with small molecules[6,7]. Small molecules are capable of recognizing these three-dimensional pockets by a mechanism that is distinct from sequence-based recognition[8]. Interactions between RNAs and protein or small molecule ligands can influence the RNA conformational ensembles by stabilization or alteration of populations of structures, leading to modified function[9]. Changes in structure-dependent regulatory processes are key to normal function and can lead to the manifestation of various disease states[10,11]. Understanding how RNA structures respond to interacting partners is valuable[12], as this can be leveraged for drug development as well as a better understanding of endogenous gene regulation with small ligands[13,14].

Riboswitches are intriguing systems for investigating mechanistic aspects of RNA-ligand interactions, where well-defined and complex three-dimensional folds have evolved to enable small molecule recognition and altered gene expression[15,16]. Riboswitches regulate gene expression specifically through ligand binding to three-dimensional folded structures, primarily in bacteria[17]. One class of commonly identified riboswitches are responsible for sensing the PreQ$_1$ metabolite (7-aminomethyl-7-deazaguanine). These riboswitches are associated with the regulation of levels of hypermodified guanine, prequeuosine, and queuosine itself, a metabolite used in the posttranslational modification of tRNAs[18]. Evolutionarily diverse PreQ$_1$ riboswitches encompass three distinct classes—Class I, II, and III— and differ fundamentally in structure, ligand binding, and gene regulatory mechanisms[19]. Class I (PreQ$_1$-I) riboswitches feature a singular aptamer domain that directly interacts with PreQ$_1$, initiating transcriptional attenuation or termination and, in some cases, inhibition of translational initiation[20,21]. Class II (PreQ$_1$-II) riboswitches are characterized by a pseudoknot structure formed by the aptamer domain, enabling ligand recognition and direct gene regulation[22,23]. Notably, Class III (PreQ$_1$-III) riboswitches possess a distinct architecture, with the aptamer and expression platforms segregated in sequence. This class modulates gene expression through conformational changes upon PreQ$_1$ binding[24]. These variations highlight the multifaceted mechanisms employed by PreQ$_1$ riboswitches in bacteria to dynamically regulate gene expression in response to ligand binding[25]. Riboswitches have long been considered compelling targets for small molecules. Early studies focused on medicinal chemistry efforts to develop synthetic compounds derived from riboswitches, including examples such as FMN[26,27], TPP[28,29], glmS[30–32] PreQ$_1$[33,34], and lysine[35]. In addition, riboswitches represent rare cases where atomic resolution structures of small molecules in complex with RNA have been solved, enabling detailed biophysical analysis. Previously, our laboratory has studied multiple riboswitches as targets, including the ZMP riboswitch[36] and multiple PreQ$_1$ riboswitches[37,38].

Here, we present a structure-informed approach to developing biologically active ligands for class I PreQ$_1$ (PreQ$_1$-I) riboswitches that involve modifying a chemical scaffold to enable biological activity. We report multiple synthetic ligands, including 4, that can directly bind to the PreQ$_1$ riboswitch despite no obvious chemical similarity to PreQ$_1$ itself. Along with other derivatives, 4 displayed tight binding affinity to the *Bacillus subtilis* (*Bsu*)-PreQ$_1$ riboswitch. Investigation of five other PreQ$_1$ riboswitches diverse in sequence, structure, function, and evolutionary origin revealed that this synthetic ligand exhibited similar conformational effects to PreQ$_1$ in most cases. An X-ray co-crystal structure of the ligand in complex with a PreQ$_1$ riboswitch revealed an identical binding site, but distinct binding mode relative to PreQ$_1$. A second scaffold, based on the harmol heterocycle (8), was also co-crystallized with the aptamer, and evaluated in functional assays. This compound displayed in vitro activity but was inactive in cell-based assays, potentially due to the more promiscuous nature of the scaffold interacting with other RNAs. Single-molecule assays revealed that PreQ$_1$ induces stable pseudoknot formation. However, the binding of 4

impacts riboswitch function by a distinct kinetic mechanism, altering the rate of folding and most likely stabilizing the PreQ$_1$ RNA in a partially folded "pre-pseudoknot" state, despite having the same conformational consequence in bulk measurements. Both in vitro, transcription termination, and in vivo expression assays in bacterial cells validate the ability of 4 to impact gene expression by binding directly to RNA. This work demonstrates that diverse chemical scaffolds can bind to and influence riboswitch aptamers to accomplish similar functional outcomes through distinct mechanisms.

## Results

### Structure-informed alterations in chemical structure impact binding to aptamer

Previous work demonstrated that a synthetic dibenzofuran ligand has high affinity and selectivity to both *Bsu*-PreQ$_1$ and *Tte*-PreQ$_1$ riboswitches[37]. We used ICM MolSoft software[39] to dock various chemical scaffolds related to the initial dibenzofuran hit compound that had been reported previously. We conducted structural modifications and synthesized various xanthone derivatives to assess their biologically relevant interactions with the PreQ$_1$ riboswitch. By altering the side chains and incorporating other modifications, as detailed in Fig. 1A, we synthesized several analogs and examined their recognition ability and activity. A goal of this exercise was to identify synthetically accessible chemical scaffolds potentially capable of improved affinity or activity by making more contact with the RNA. Using this approach, we designed and synthesized nine small molecule ligands representing heterocyclic cores or sidechains that could plausibly bind to the PreQ$_1$ aptamer.

To evaluate the binding of each compound to the RNA, we employed fluorescence titrations or microscale thermophoresis (MST)[40]. Compounds 1, 2, 4, 5, and 8 showed changes in ligand fluorescence with increasing concentrations of RNA. For these compounds, ligand fluorescence was plotted as a function of RNA concentration. Data were fitted using a one-site total binding model to measure an approximate equilibrium dissociation constant ($K_D$) for each of these compounds. Next, compounds that did not show any fluctuation of ligand fluorescence, (3, 6, 7, and 9) were evaluated using MST using a Cy5-labeled *Bsu* PreQ$_1$ aptamer. By fitting the curves as a function of ligand concentration and using a one-site total binding model, an approximate equilibrium dissociation constant ($K_D$) was measured (Fig. 1A and Supplementary Fig. S1). In general, most of the ligands bound to the RNA with low micromolar affinity. However, compound 9 showed no binding up to a concentration of 500 μM. Of the remaining compounds, 4 showed micromolar binding (approximate $K_D = 16 \pm 21$ μM), and therefore binding was also evaluated with another dye, AlexaFluor 647, to rule out potential effects due to the fluorophore used in binding analysis. Using labeled aptamers from *Staphylococcus saprophyticus* (*Ssa*)-PreQ$_1$ and *Tte*-PreQ$_1$ that have a conserved binding domain, compound 4 demonstrated approximate $K_D$ values of $21.9 \pm 2.25$ μM and $29.0 \pm 2.4$ μM, respectively, in FIA (Supplementary Fig. S2), confirming direct binding to the RNA. For reference, the affinity of cognate ligand PreQ$_1$ with *Bsu* PreQ$_1$ aptamer, as previously determined, is $4.1 \pm 0.6$ nM[37]. Binding assays were also performed with compounds 4 and 8 with six diverse RNA sequences that can form various RNA secondary structures, to establish specificity (Supplementary Fig. S3). Here, 4 and 8 show no apparent binding with these structures. Next, compounds were used in in vitro assays for functional evaluation.

### Synthetic ligands are active in single-round transcriptional termination assays

In vitro, single-round transcription termination assays were performed to biochemically analyze the activity of analogs in functional assays. The *Ssa*-PreQ$_1$ riboswitch was subjected to transcription in the presence of increasing concentrations of ligands, which could either result

## (A)

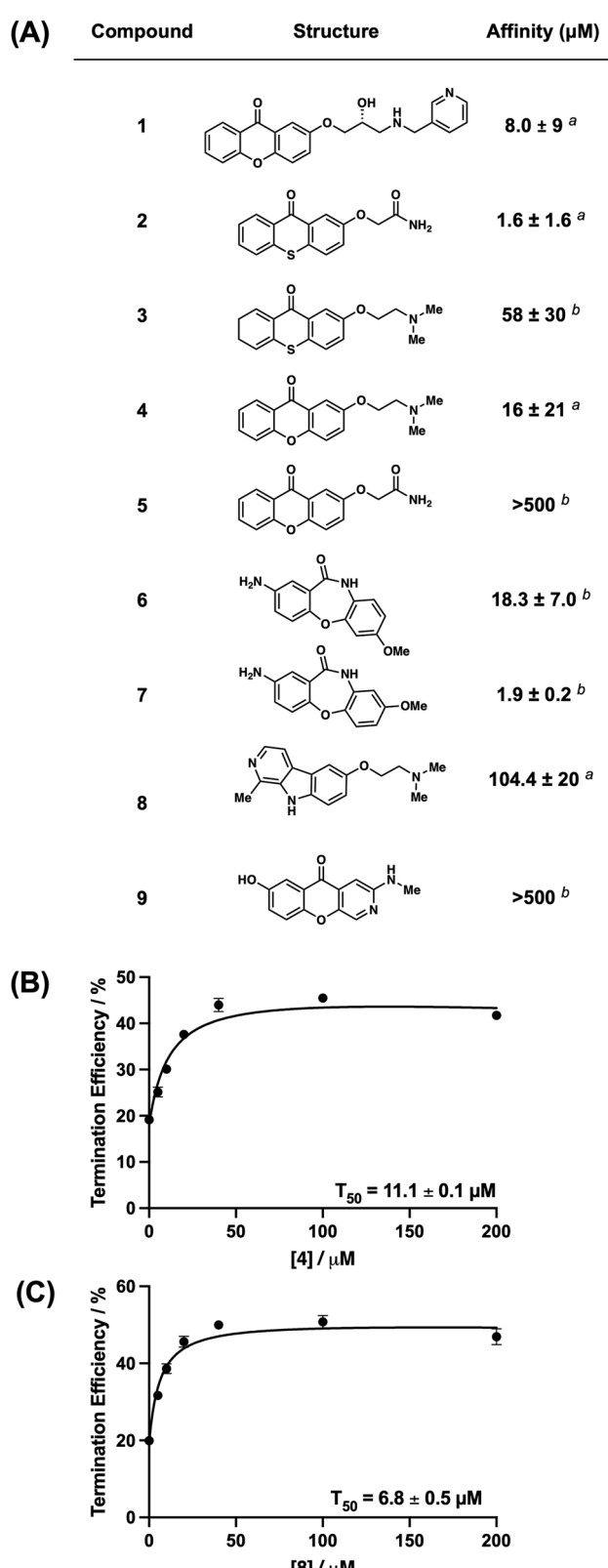

| Compound | Structure | Affinity (µM) |
|---|---|---|
| 1 | | $8.0 \pm 9$ [a] |
| 2 | | $1.6 \pm 1.6$ [a] |
| 3 | | $58 \pm 30$ [b] |
| 4 | | $16 \pm 21$ [a] |
| 5 | | $>500$ [b] |
| 6 | | $18.3 \pm 7.0$ [b] |
| 7 | | $1.9 \pm 0.2$ [b] |
| 8 | | $104.4 \pm 20$ [a] |
| 9 | | $>500$ [b] |

## (B)

$T_{50} = 11.1 \pm 0.1 \, \mu M$

## (C)

$T_{50} = 6.8 \pm 0.5 \, \mu M$

**Fig. 1 | Structure, binding measurements, and activity of synthetic ligands.**
**A** Chemical structures of PreQ$_1$ riboswitch aptamer ligands and their approximate binding affinities to a *Bsu*-PreQ$_1$ aptamer. [a] is K$_D$ measured using intrinsic ligand fluorescence, [b] is K$_D$ measured by MST measurements, **B**, **C** Quantification of transcription termination efficiencies (T$_{50}$ values) of **4** and **8**, respectively, as a function of concentration of each of the ligand in single round transcription termination assays. Error bars on T$_{50}$ values represent the standard deviation from three replicate measurements.

in a full transcript read-through (RT) or a transcription termination due to the formation of terminator hairpin (T). T and RT products were visualized on a denaturing PAGE, and the termination efficiency (T$_{50}$) value was calculated by dividing the terminated band intensity by the total RNA intensity. These data show enhanced biochemical activity of **4** (T$_{50}$ = 11.1 ± 0.10 µM) and **8** (T$_{50}$ = 6.8 ± 0.45 µM) in vitro relative to the initial dibenzofuran. (Fig. 1B, C and Supplementary Fig. S4). Although **8** had weaker binding, the reason for the increased activity of this compound in transcriptional termination assays remains unclear. For the remaining compounds, saturation was not observed at the limit of solubility, and therefore accurate T$_{50}$ values could not be measured. Since compounds **4** and **8** showed activity in functional assays, they were studied further.

### X-ray co-crystal structure establishes ligand binding mode

To further understand the binding mode of **4** and **8**, we performed X-ray crystallography on the ligand aptamer complex. The co-crystal structures of the abasic mutant at positions 13, 14, and 15 in *Tte*-PreQ$_1$ riboswitch aptamer (ab13_14_15) with **4** and **8**, were determined at 2.15 Å and 2.25 Å resolution, respectively, by molecular replacement method (Fig. 2A, B and Supplementary Table S4). Compound **4** binds at the PreQ$_1$ binding site, where the xanthone core is sandwiched by one face with G11 and the other with G5 and C16, residues that are strictly conserved in the class I PreQ$_1$ riboswitches. When the current co-crystal structure is superimposed onto the PreQ$_1$-bound form, the planar rings of their ligands are well overlapped (Fig. 2C). However, because **4** is bulkier than PreQ$_1$ and its heteroatom content is less than that of PreQ$_1$, the binding pose of **4** slightly diverges from that of PreQ$_1$. In the co-crystal structure with PreQ$_1$, one side of the base containing the N2, N3, and N9 atoms of PreQ$_1$ is recognized by strictly conserved hydrogen bonds with the N1 and N6 atoms of A29 and the O4 atom of U6 of the *Tte*-PreQ$_1$ riboswitch, respectively (Fig. 2D). In contrast, the corresponding side of the xanthone moiety of **4** is further from these crucial atoms, resulting in a tilted binding axis of the heterocyclic core of **4** compared to PreQ$_1$ of approximately 15 degrees (Fig. 2C). Consequently, the oxygen atom of the central ring of **4** is situated at 3.63 Å away from the N6 atom of the phylogenetically conserved A29 (Fig. 2E). This finding suggests a weak hydrogen bonding interaction between the riboswitch and compound **4**, unlike the strong interaction observed in the PreQ$_1$-bound form where the distance between the N6 atom of A29 and the N3 atom of PreQ$_1$, a counterpart of the oxygen atom of the central pyranoid ring of **4**, is 3.13 Å. In addition, the carbonyl oxygen atom of 4 and 2'-hydroxyl group of G11 of the riboswitch seem to form a weak hydrogen bond (interatomic distance is 3.75 Å). The superimposition of these two structures indicates that **4** collides with the base of C15 of the PreQ$_1$-bound structure, due to the size of **4** being larger than that of PreQ$_1$. Consequently, the conformation of the sugar and phosphate backbone at position 15 of the current structure is relocated to fit **4** into the ligand binding site, when compared to the PreQ$_1$-bound form.

Like **4**, compound **8** is situated in the PreQ$_1$ binding site and is surrounded by the phylogenetically conserved nucleotides (Fig. 2F). Similar to the carbonyl oxygen atom of 4, the amine of the pyrrole ring of 8 is in a weak hydrogen bonding distance (3.66 Å) with 2'-OH group of G11. Since the interatomic distance is too far to interact tightly and there is no other hydrogen bonding interaction between 8 and the riboswitch, 8 is primarily stabilized in the ligand binding site of the riboswitch by stacking and hydrophobic interactions. While **4** forms a hydrogen bond with the riboswitch, **8** does not hydrogen bond with any nucleotides. Therefore, **8** is stabilized in the ligand binding site of the riboswitch by stacking and hydrophobic interactions. Compared to the binding site of **4**, the binding site of **8** is shifted about 1-2 Å in the opposite direction of the L2 loop. This is likely because **8** has only one weak hydrogen bond with the riboswitch, which makes it possible to

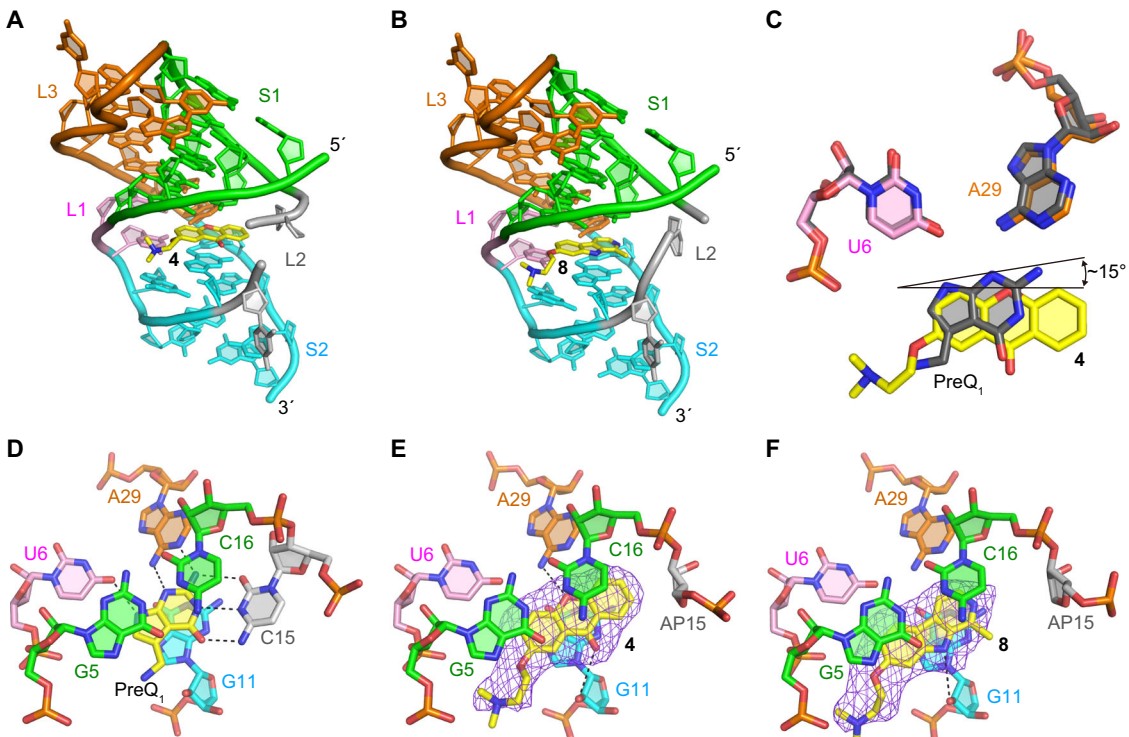

**Fig. 2 | X-ray crystal structures of ab13_14_15 in complex with synthetic ligands.** **A**, **B** Overall structure of the complexes with ligands **4** (**A**) and **8** (middle). **C** Comparison of binding poses between **4** and PreQ$_1$. **D** Structural comparison between the wild-type *Tte*-PreQ$_1$ riboswitch aptamer complexed with PreQ$_1$ (PDB ID: 3Q50)[70], (**E**) ab13_14_15 complexed with **4** and (**F**) ab13_14_15 in complex with **8**. Hydrogen bonds are shown in dashed lines. Purple mesh represents the m$F_o$-D$F_c$

electron density maps observed for each ligand, which are contoured at 2.5 σ. The compounds were omitted from the phase calculation. The distances between the oxygen atom of the central ring of **4** and the N6 atom of A29, the carbonyl group of **4** and 2′-OH group of G11, and the amine of the pyrrole ring of **8** and 2′-hydroxyl group of G11 are 3.63, 3.75, and 3.66 Å, respectively.

shift the ligand binding site flexibly (Fig. 2F). In our previous report[37], we analyzed the effects of dibenzofuran and carbazole derivatives on PreQ$_1$ riboswitch function and showed that the binding poses of these compounds differ due to changes in the acceptor/donor pair of hydrogen-bond between these compounds and the riboswitch. Ligand **8** is a derivative of harmol and has a nitrogen atom in the central ring like the previous carbazole derivative. However, the binding pose of **8** is quite different when compared to other nitrogenous heterocycle ligands (such as PDB ID: 6E1V)[37], and the nitrogen atom of the central ring of **8** faces in the opposite direction. Therefore, the conserved nucleotides, U6 and A29, that are crucial for recognizing PreQ$_1$ by hydrogen-bonds only contact with the heterocycle of **8** via van der Waals interactions. Together, these structures provide a rationale for both how diverse ligands recognize the aptamer binding site and why they are active in functional assays. It is important to note that C15 of the riboswitch is critical for recognizing PreQ$_1$ via the canonical Watson-Crick base pairing. Therefore, the binding of **4** and **8** to the PreQ$_1$ binding site would affect the conformations of L2 and S2, which are important for regulating the riboswitch function. Consistent with this, the co-crystal structures with **4** and **8** exhibit conformational differences of L2 and S2 when compared to those in the PreQ$_1$-bound form. In the PreQ$_1$-free structure, the conserved A14 occupies the PreQ$_1$ binding pocket. When PreQ$_1$ is bound to the riboswitch, the conformation of L2 is restructured to acquire the space for the ligand binding, resulting in the recognition of PreQ$_1$ by C15. In addition, together with S2, the nucleotides of L2, including A13, A14, and C15, form a nucleobase-stacking spine that is crucial for controlling gene regulation[41]. Since the abasic mutant at positions 13, 14, and 15 was used in this study, we cannot rule out the possibility that the conformational differences are due to the introduction of the abasic sites in the current construct. However, given the steric hindrance between

C15 and compounds **4** and **8**, these compounds probably have a major effect on the structure of these regions. Both L2 loops in the co-crystal structures with **4** and **8** seem to adopt an intermediate-like conformation between the PreQ$_1$-free and PreQ$_1$-bound states (Fig. 3). Because of the absence of the nucleobases, ab13_14_15 does not form the nucleobase-stacking spine in the current co-crystal structures. Consequently, the conformation of S2 is similar to that of the PreQ$_1$-free state. It is supposed that the wild-type *Tte*-PreQ$_1$ riboswitch forms the nucleobase-stacking spine through the interactions with **4** and **8** by adopting the PreQ$_1$-bound-like structure, despite that these compounds are bulkier than the cognate ligand, which could be related to the differences in the results of biochemical analyses described below.

## Structure probing reveals impacts of ligand binding on aptamer flexibility

PreQ$_1$ riboswitches are among the most commonly evolved riboswitches, and as such, have been observed to have considerable diversity in terms of sequence, structure, and mechanisms[25]. Given the diversity of RNA structures that recognize the PreQ$_1$ metabolite, we asked whether evolutionarily diverse PreQ$_1$ aptamers have differential effects on ligand-mediated recognition and flexibility. We utilized selective 2′-hydroxyl acylation analyzed by primer extension and mutational profiling (SHAPE-MaP) to assess the flexibility of bases at single nucleotide level in the presence and absence of both PreQ$_1$ and **4**[42]. PreQ$_1$ RNA aptamers from six different species were selected, representing all three classes of PreQ$_1$ riboswitch. We studied aptamers from *Tte*[19,43], *Bsu*[19,43], and *Ssa*[37] aptamers from PreQ$_1$-I, *Lactobacillus rhamnosus (Lrh)*[43–45], and *Streptococcus pneumoniae (Spn)*[46,47] from PreQ$_1$-II and *Faecalibacterium prausnitzii (Fpr)*[24] belonging to PreQ$_1$-III riboswitches.

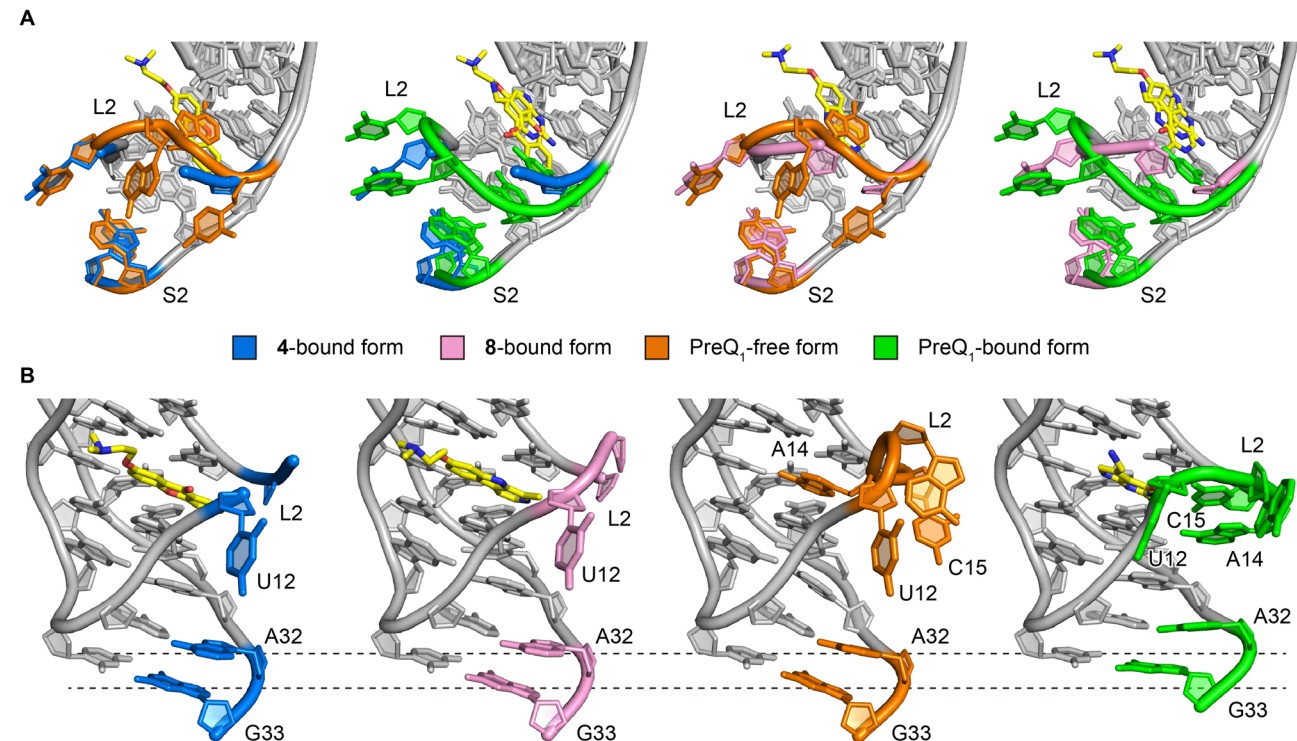

**Fig. 3 | L2 and S2 structures of the PreQ₁-free, PreQ₁-bound, and the compounds 4 and 8-bound forms. A** Superimposed structures of the **4**-bound and PreQ₁-free forms (left), the **4**-bound and PreQ₁-bound forms (second left), the **8**-bound and PreQ₁-free forms (second right), and the **8**-bound and PreQ₁-bound forms (right). L2, A32, and G33 in S2 are color-coded as indicated. PreQ₁ and compounds **4** and **8** are in yellow. **B** Structural comparison highlighting the conformational differences of A32 and G33 in S2, with the same color codes as (**A**).

Each in vitro synthesized RNA was folded and incubated with DMSO, PreQ₁, or **4**, followed by incubation with the SHAPE reagent 2A3[48]. Modified and unmodified RNAs were reverse transcribed, and mutations were mapped by next-generation sequencing. Data analysis using the Shapemapper pipeline[49] revealed mutation rates and the reactivity profile for each nucleotide. Here, lower SHAPE reactivity depicts decreased flexibility (or stabilization) of each nucleoside in the presence of a ligand. Next, SHAPE constraints for each nucleotide were utilized to predict secondary structure with RiboSketch software[50]. To better represent complex interactions found in pseudoknots, the SHAPE-derived constraints were superimposed on the secondary structures informed by x-ray co-crystallography or NMR-derived conformations previously (Fig. 4A−F). Base-pairing probabilities using the SHAPE-derived data are shown using the arc plots using Superfold[51]. Delta SHAPE analysis was then used to identify nucleotides specifically altered in flexibility upon binding to the ligand[52]. After accessing significant changes in the SHAPE reactivity within different riboswitches, *Bsu, Lrh, Spn, and Fpr* showed alteration of structure in the presence of both ligands. Specifically, the *Bsu* riboswitch showed stabilization of structure at C15, A16, and C17 belonging to aptamer domain[20], as well as at the C48 and U55 bases within the terminator hairpin domain. This result could reflect the observation that the inclusion of a terminator hairpin in the aptamer construct enables the folding of a more stable pseudoknot in an NMR study[53]. A37, C38, G39, and terminator hairpin bases U50, U51, and G52 display destabilization in the presence of PreQ₁ ligand, demonstrating increased solvent exposure due to non-participation in base-pairing and remain single-stranded (Supplementary Fig. S5A). In the presence of **4**, there was no significant change observed with delta SHAPE. With PreQ₁ bound to *Spn* aptamer, A52, G53, G54, A55, G56 (belonging to the loop J2-4) were stabilized and A41, U42, A43, A44, C45 (P4 stem)[46] are destabilized. This effect was strikingly similar in the presence of **4**, as A52, G53, G54, and A55 were stabilized along with the destabilization of A41, U42, A43, and A44

bases (Supplementary Fig. S5B, C). The *Lrh* aptamer in presence of PreQ₁, A34, U35, U36, C39, U40, U41 (J2-3 loop), G60 (P4 region) were observed to have positive delta SHAPE inferring stabilization and bases U52, A53, U 54, U55, A56 (J2-4 loop), A62, A63 (P4 region)[23] had negative delta SHAPE. With **4**, the effect was similar to PreQ₁, where bases U33, A34, U35, U36, C39, U40 along with U52, A53 displayed stabilization in the aptamer. However, bases G43, A44, U45 (P3), U55, A56 (J2-4), A72, G73, G74, and A75, incorporated in the ribosome binding site (RBS) showed significant destabilization (Supplementary Fig. S5D, E). In *Fpr* PreQ₁-riboswitch, with delta SHAPE, destabilizing events were captured with both ligands, as bases G79, G80, A81, and G82 (constituting ribosome binding site) had enhanced reactivities. With PreQ₁, only A84 showed positive delta SHAPE, meaning stabilization (Supplementary Fig. S5F, G). In contrast, the *Tte* PreQ₁ riboswitch displayed only destabilization in the presence of PreQ₁ (at A24, C25, A26, A27, A28, and A29, which have no interaction with the PreQ₁ ligand) and had no significant delta SHAPE reactivity when **4** was bound (Supplementary Fig. S5H). In general, all structures displayed decreased reactivity in the presence of both ligands in comparison to the DMSO control, reflecting an overall stabilization of the structure. Both natural (PreQ₁) and synthetic ligands (**4**) had strikingly similar effects on RNA conformation except for a few nucleotides. While these secondary structures are experimentally informed, they do not necessarily reflect three-dimensional aptamer structures with perfect accuracy. Still, this comparative analysis is a powerful demonstration that chemically distinct ligands can have similar effects on structurally diverse RNAs that recognize a common cognate ligand.

## Ligand binding stabilizes RNA structure

Having confirmed that ligand binding leads to significant changes in the structure of PreQ₁ RNA using bulk methods in solution, we next used Depixus' magnetic force spectroscopy (MFS) platform to evaluate the effects of ligand binding on the stability of the aptamer's

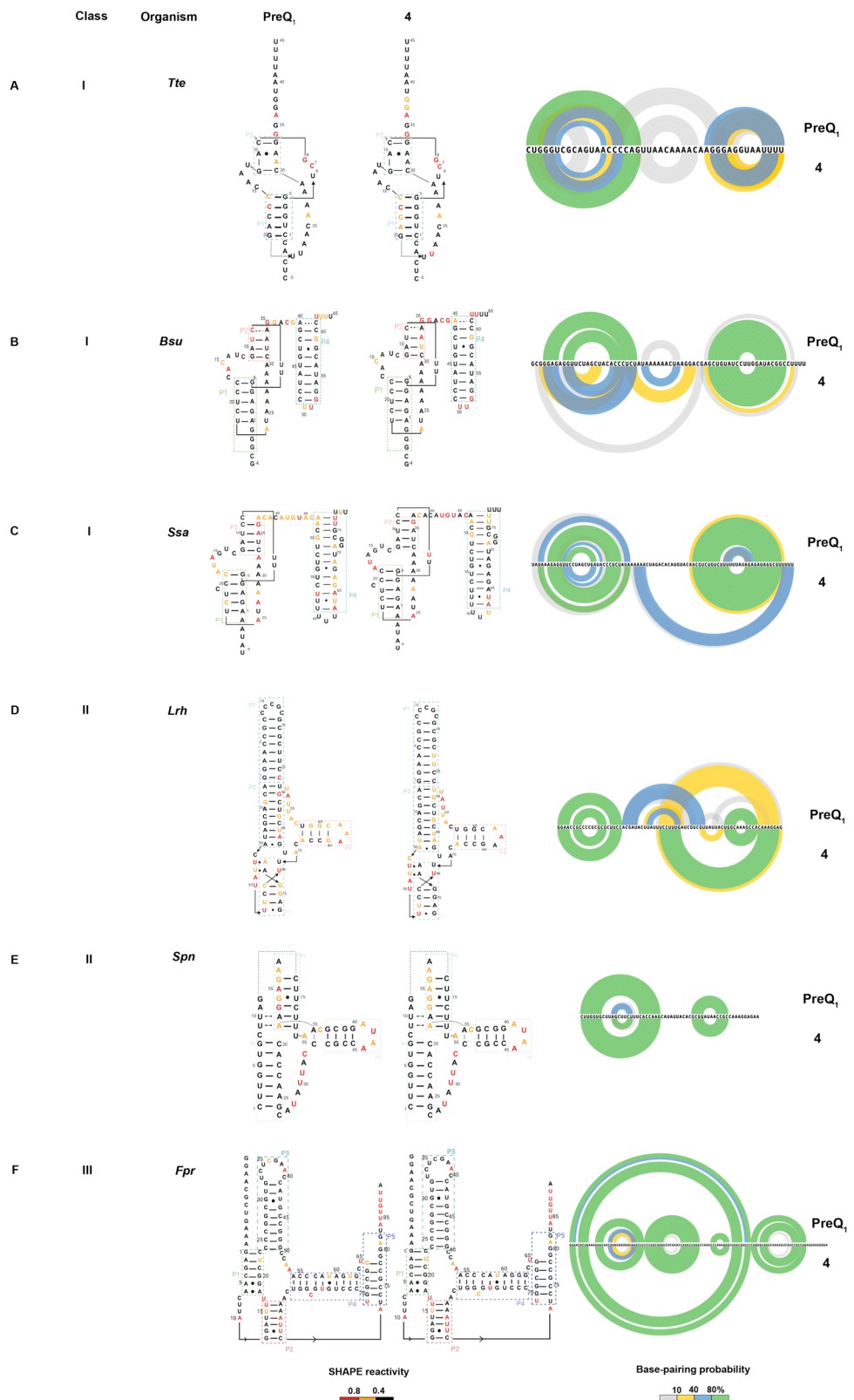

**Fig. 4 | Structural changes in RNA upon ligand binding. A–F** SHAPE-MaP informed secondary structure predictions of various riboswitches belonging to three evolutionarily different PreQ$_1$ riboswitch classes in the presence of cognate (PreQ$_1$) (left) and synthetic ligand (**4**) (middle). Shown are the representative predictions from three technical replicates. Arc plots on the right show the base pairing probability of each riboswitch in the presence of PreQ$_1$ and **4**. SHAPE reactivity and base pairing probabilities are indicated using the respective color schemes shown at the bottom.

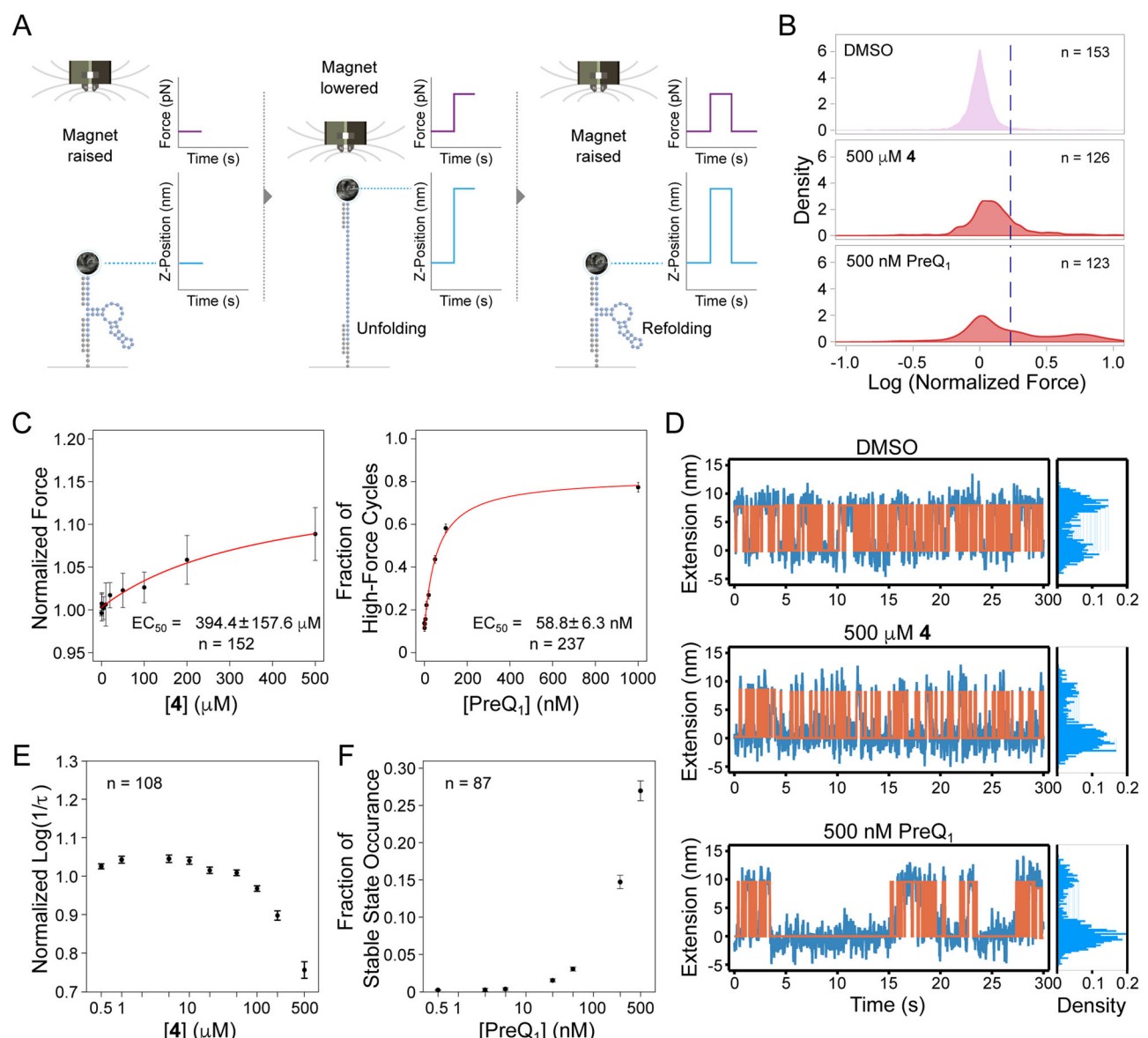

**Fig. 5 | Magnetic force spectroscopy (MFS) analysis of ligand binding.**
**A** Overview of Depixus' single-molecule MSF platform for exploring the interactions of bioactive small molecule ligands with their target RNA structures in real-time. **B** Unfolding force distributions of the *Bsu* PreQ$_1$ riboswitch aptamer in control, **4** and PreQ$_1$ ligand conditions. **C** Dose-response curve for the change in unfolding of the aptamer in the presence of **4** and PreQ$_1$. **D** Raw traces of the constant-force experiments for control, **4**, and PreQ$_1$ of a single molecule with cumulative density histograms are shown to the right. **E** The aptamer unfolding rate as a function of the concentration of **4** in constant force experiments. **F** The impact of PreQ$_1$ ligand concentration on the occurrence probability of the stable folded state. n is the number of molecules analyzed, and the values represent mean ± standard deviation (SD).

structure at the single-molecule level. This platform allows precise tracking of molecular extension in response to an applied force across hundreds of single molecules in parallel to gain insights into molecular dynamics and interactions. To use MSF, first, a biotinylated *Bsu*-PreQ$_1$ aptamer was bound to a streptavidin paramagnetic bead and tethered to a flow cell floor via hybridization to a surface-bound oligonucleotide. A precisely controllable magnetic force was then applied to the beads whilst their vertical or Z-positions were tracked in real-time. Any clumped or close-proximity beads were excluded from the analysis to prevent interference between the molecules or beads. When the RNA was subjected to low force, it folded freely. As the force was increased, structural disruption or unfolding occurred, resulting in a sudden change in vertical bead position. The force could then be reduced, allowing the structures to return to a folded conformation (Fig. 5A). This non-destructive process was repeated over multiple cycles of slowly increasing, then decreasing forces (referred to as force ramp

experiments, Supplementary Fig. S6A), while the forces at which individual structures unfolded and refolded were measured. The addition of ligands to the flow cell allowed tracking of their impact on the stability of the RNA structures through their effect on these unfolding and folding forces. Separately, stepped constant-force experiments were performed where RNA molecules were subjected to the same force for a fixed amount of time before increasing the force in a stepwise manner (Supplementary Fig. S6A). During each forced step, the transition of the RNA between the unfolded and folded states was tracked, and the time spent in the unfolded state was observed to increase with force until the RNA structures remained constantly unfolded. The equilibrium force at which the RNA spent equal time in each state was also determined. Constant force experiments in which the RNA was subjected to the equilibrium force for an extended period could then be performed, to allow the impact of ligand binding on folding and unfolding dynamics to be explored

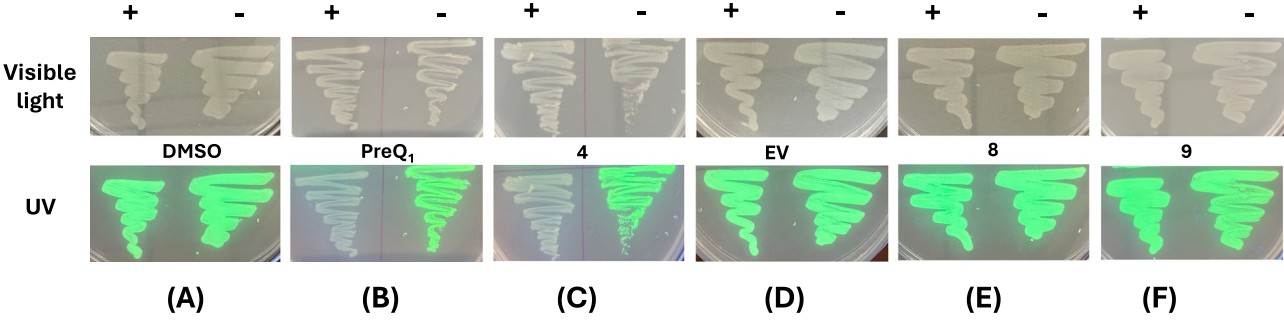

**Fig. 6 | Synthetic ligands impact gene expression in bacteria.** Ligands impact expression of a GFP reporter gene containing a PreQ$_1$ aptamer in mutant *E.coli* grown on specialized media in the presence and absence of DMSO (**A**), PreQ$_1$ (**B**), **4** (**C**, **D**), **8** (**E**), and **9** as negative control (**F**) visualized under visible light (top) and UV transilluminator (bottom). EV: empty vector that lacks any riboswitch construct but contains GFP. Representative images from three independent replicates are shown.

through changes in the equilibrium force and/or the frequency of folding-unfolding events (Supplementary Fig. S6A).

We conducted ramp experiments to probe RNA structure unfolding under varied conditions: control (1% DMSO), **4**, and PreQ$_1$ and plotted the distribution of the normalized forces required to unfold and refold the RNA structures. In the control condition, the force distribution formed a single peak (Fig. 5B), which was attributed to the unfolding of a "pre-pseudoknot" structure. The introduction of a saturating concentration (500 μM) of **4** subtly shifted the peak of the force distribution toward higher forces, implying minor structural influence that increases the force needed to unfold and refold the structure (Fig. 5B and Supplementary Fig. S6B). In contrast, saturating concentrations of PreQ$_1$ (500 nM) induced a second peak in higher forces, indicating that the molecules sometimes required a much higher force to unfold, which was attributed to the formation of stable pseudoknot structures. However, the position of the first peak did not shift, demonstrating that PreQ$_1$ did not change the stability of the pre-pseudoknot structure (Fig. 5B and Supplementary Fig. S6B. The second high force peak was notably absent with **4**, highlighting that the compound did not trigger the formation of persistent/stable pseudoknots like those induced by PreQ$_1$ (Fig. 5B). Importantly, this remains the case even at high ligand concentrations, further emphasizing that this effect is a mechanistic distinction rather than a concentration or K$_D$ effect (Supplementary Fig. S7).

The increase caused by compound **4** on both the median normalized unfolding and refolding forces was shown to be concentration dependent (Fig. 5C and Supplementary Fig. S6C, respectively) with EC$_{50}$ values of $394 \pm 157$ μM and $456 \pm 258$ μM respectively, suggesting that **4** stabilizes the riboswitch pre-pseudoknot structure and that this interaction helps to refold the RNA. For the PreQ$_1$ ligand, the concentration effect was assessed differently, using the fraction of high force cycles, to account for cycles in which the RNA in pseudoknot conformation did not unfold at the maximal force applied. PreQ$_1$ binding increased the fraction of the high-force cycles in a concentration-dependent manner, indicating an increase in pseudoknot formation, whilst RNA refolding was not affected by the cognate ligand (Fig. 5C and Supplementary Fig. S6D).

Under constant-force experiments, the bead position tracking of individual molecules showed that the PreQ$_1$ ligand prolonged folded state duration compared to the control and **4** (at the same applied forces), revealing ligand-induced pseudoknot formation (Fig. 5D and Supplementary Fig. S6E). Analysis of the lifetimes of the folded states under control, PreQ$_1$ (500 nM), and **4** (500 μM) conditions showed an exponential distribution of the observed events, with **4** inducing a slightly increased lifetime compared to the control. In the presence of the cognate ligand, the folded states fitted a combination of two distinct lifetime distributions (confirmed using the Bayesian information criterion) (Supplementary Fig. S6F, representing one single molecule). Of these two lifetimes, the shorter of the two showed a lifetime similar

to that of the control, most likely corresponding to the pre-pseudoknot state, and the second corresponded to the stable folded state attributed to the pseudoknot.

To evaluate the concentration dependency of the effects, lifetime data from multiple molecules were aggregated by evaluating log (1/lifetime) and normalizing each condition to the control for the same molecule before combining data from multiple molecules. Compound **4** was confirmed to decrease the unfolding rate (i.e., to cause the RNA to stay longer in the folded pre-pseudoknot state) in a concentration-dependent manner, but only at concentrations above 100 μM (Fig. 5E). In contrast, the cognate ligand did not affect the unfolding rate of either the short lifetime form (pre-pseudoknot) or long lifetime form (pseudoknot) (Supplementary Fig. S6G) and no change in the refolding rate (Supplementary Fig. S6H). Instead, the probability of the pseudoknot state occurring increased with PreQ$_1$ ligand concentration (Fig. 5F), confirming that binding of the PreQ$_1$ ligand induces pseudoknot formation in a concentration-dependent manner. Compound **4** was also demonstrated to increase the refolding rate in a concentration-dependent manner above 100 μM, suggesting that the molecule alters the rate of refolding of the RNA, perhaps by binding to a less folded form (Supplementary Fig. S6I). However, while the PreQ$_1$ ligand affects the rate of pseudoknot formation, it has no effect on the less stable pre-pseudoknot structure's folding dynamics.

## Ligands affect riboswitch activity in cells

Next, we evaluated the ability of **4** to modulate riboswitch activity in vivo. We employed an engineered green fluorescent protein (GFPuv) reporter assay, as has been used previously to demonstrate riboswitch activity[54,55]. The *Bsu*-PreQ$_1$ riboswitch aptamer was cloned into a plasmid bearing GFPuv expression in parallel with a second, empty vector that expresses GFP but lacks any riboswitch. Next, GFPuv positive constructs were transformed into the JW2765 strain of *E.coli* bearing a *ΔqueF* mutation[56,57] to generate a stable cell line expressing the reporter construct. The *ΔqueF* mutation leads to impaired PreQ$_1$ biosynthesis and provides an ideal system to study the effects of ligands on the PreQ$_1$ riboswitch, as it lacks endogenous PreQ$_1$. Next, cells were grown on specialized CSB media (to further hinder any endogenous PreQ$_1$ biosynthesis) in the presence of compounds or the DMSO control. When visualized under UV, cells grown in the presence of DMSO exhibited high levels of fluorescence (Fig. 6A). In contrast, treatment with PreQ$_1$ and **4** led to a complete loss of fluorescence levels (Fig. 6B, C). The cell line expressing an empty vector was not responsive to ligand (Fig. 6D). In addition, a compound structurally similar to **4** that did not bind to the riboswitch (compound **9**) was also inactive. Finally, cells treated with **8** also did not respond, even though **8** both binds to and modulates the function of the riboswitch in vitro. Importantly, these results demonstrate that both PreQ$_1$ and **4** most likely exhibit gene modulation activity by directly binding to RNA

structures in cells, rather than nonspecific or other off-target mechanisms.

## Discussion

In this work, we demonstrate that structure-informed design can be used to identify small molecules that bind to RNA structures and impact their function in biophysical, biochemical, and biological assays. By investigating diverse heterocyclic scaffolds, we identified compounds with considerably improved activity in single-round transcription termination assays. Here, subtle changes in compound structure can have dramatic impacts on both binding affinity and activity in functional assays, resulting in a complex and non-obvious structure-activity relationship (SAR). X-ray crystallographic analyses indicate that multiple scaffolds can bind to the $PreQ_1$ aptamer RNA at the same binding site as the cognate ligand with well-defined binding modes. Further, some ligands exhibited additional specific bonding interactions with the RNA.

In addition to binding assays, SHAPE-MaP studies were used to observe ligand-induced effects on the stability of six evolutionarily and functionally diverse $PreQ_1$ riboswitches, all of which recognize the same cognate ligand. In each instance, the synthetic ligand exhibited effects similar to the native ligand. This was demonstrated through similar alterations in base-pairing interactions contributing to the stabilization of the riboswitch structure despite the lower binding affinity of the synthetic ligand. Thus, although these RNAs represent distinct sequences, are structurally diverse, and exhibit switching activity through different mechanisms, they recognize small molecules through conserved three-dimensional structures rather than sequences.

In addition, single-molecule magnetic force spectroscopy was used to assess the ability of $PreQ_1$ and **4** to impact aptamer folding. Here, the presence of cognate ligand led to the formation of a stable pseudoknot structure. In contrast, the binding of **4** appears to have a distinct mechanism. These experiments indicate that **4** stabilizes the $PreQ_1$ RNA, most likely in the pre-folded pseudoknot state. Thus, **4** has a different impact on the folding pathway compared with the cognate $PreQ_1$ ligand, impacting the rate of re-folding rather than inducing a stable pseudoknot formation. More specifically, the activity observed by the recognition of **4** is possibly due to kinetic alterations rather than thermodynamic factors, as with $PreQ_1$. Riboswitches are known to exhibit diverse folding pathways and mechanisms of switching, reflecting a complex interplay between ligand binding and structural dynamics that results in altered gene expression. Although, in many cases, persistent formation of a folded state is thought to be required, it has also been observed that alteration of folding kinetics can drive switching behavior as seen with the ZTP riboswitch[58,59].

Finally, we used a fluorescent reporter assay to demonstrate that **4** is cell permeable and modulates riboswitch activity in live cells by directly interacting with the RNA. Importantly, both an empty vector and a chemically related, non-binding control compound showed no activity. Interestingly, the harmol-based ligand **8** displays weak binding but stronger inhibition in functional biochemical assays and was inactive in the in vivo reporter assay. While the reason for the lack of activity is unclear, it may reflect more promiscuous binding of harmol-like molecules to diverse RNA structures, which would presumably require a higher concentration to observe activity given the higher abundance of other RNAs in cells. Alternately, activity in the biochemical assay could be occurring due to nonspecific RNA binding. For example, related β-carboline alkaloids such as harmine and harmaline engage with RNA bases through interactions involving the $O^2$ of cytosine and uracil, the $N^7$ of guanine and adenine, as well as the phosphate group in the backbone, and can engage in intercalative interactions of diverse RNAs[60]. Thus, not only binding affinity and mechanism of recognition but also specificity of binding impacts RNA-ligand interactions in complex, biologically relevant settings. Taken together, these results demonstrate that even though diverse ligands can bind to the same aptamer binding site, factors including selectivity, mode of recognition, and impacts on both conformational kinetics and thermodynamics can all play a role in the ability of a compound to modulate biological function by binding to RNA.

## Methods

### Microscale thermophoresis

100 nM $Cy5$-labeled *Bsu* $PreQ_1$ (IDT) was prepared in 1X $PreQ_1$ buffer (50 mM Tris, pH 7.5, 100 mM KCl, 1 mM $MgCl_2$) and annealed by heating to 75 °C for 5 min with slow cooling to room temperature (RT) for 30 mins. Serial dilutions of the compounds were prepared in 10% DMSO in 1X $PreQ_1$ buffer. Equal volumes of folded RNA and compound were equilibrated for 15 min at RT. Using premium coated capillaries, MST was conducted in triplicate on a Monolith NT.115 system (Nano-Temper Technologies). Obtained values were plotted against the ligand concentration, and a dissociation constant ($K_D$) was determined for each of the compounds using a single-site binding model.

### Fluorescence intensity assay

$Cy5$-labeled *Bsu* $PreQ_1$ aptamer RNA construct, AF647- labeled *Tte* $PreQ_1$, and *Ssa* $PreQ_1$ RNA were purchased from IDT and annealed in $PreQ_1$ folding buffer as above. Compound dilution was prepared at a concentration range of 0 to 250 µM in 1X $PreQ_1$ buffer with 5% DMSO. RNA to a final concentration of 50 nM is added to each well and incubated at RT for 15 min. The spectrofluorometer FluoroMax-4C (Horiba Instruments) was used to read the fluorescence intensity at an excitation wavelength of 645 nm and an emission wavelength of 650–750 nm. The fluorescence intensities were plotted against ligand concentration. The binding affinity was calculated by fitting the curve using the single-site binding model in GraphPad Prism 10.

### Single-round transcription termination assay

The transcription termination assays were carried out as previously reported[37]. The DNA fragment containing λPR promoter, 26-nt C-less sequence followed by the *Ssa* $PreQ_1$ riboswitch, and its downstream sequence, was amplified by PCR from the plasmid. The PCR product was then gel-purified and used as the DNA template for transcription termination assay. Halted transcription complexes were prepared in a solution containing 1 µM GTP, 5 µM ATP, 5 µM UTP, 100 µM ApU, [α-32P] GTP, 75 nM DNA template, 0.0167 U/µL *E. coli* RNA polymerase holoenzyme (New England BioLabs) in 1X transcription buffer (20 mM Tris-HCl, pH 8.0, 2 mM NaCl, 1 mM $MgCl_2$, 4% glycerol, 0.1 mM DTT, and 0.1 mM EDTA), and incubated at 37 °C for 15 min. A DNA oligo-nucleotide complementary to the C-less sequence was added to the reactions (final concentration of 1.1 µM) in 1X transcription buffer and incubated at room temperature for 5 min. Subsequent elongation was restarted by combining 9 µL of halted transcription complex, 3 µL of each compound (0–5 mM compound and 25% DMSO in 1X transcription buffer), and 3 µL of NTPs mix (200 µM each NTP, 100 µg/mL heparin, and 250 mM KCl in 1X transcription buffer), and incubated at 37 °C for 20 min. Then, 0.5 U of RQ1 RNase-Free DNase (Promega) was added to the reactions and incubated at 37 °C for 10 min to cleave the DNA template. The reactions were stopped by adding an equal volume of loading dye (8 M urea, 20% sucrose, 0.05% bromophenol blue, and 0.05% xylene cyanol in 2X TBE). The reaction mixture was separated by 8% denaturing PAGE and visualized by phosphorimager. The band intensity was analyzed by ImageQuant software (GE Healthcare). Termination efficiency was calculated by dividing the intensity for the terminated RNA band by those for the total (terminated and antiterminated) RNAs.

### X-ray Co-crystal structure determination and refinement

Abasic mutant at positions 13, 14, and 15 (ab13_14_15) of the $PreQ_1$ riboswitch aptamer domain from *Tte* was co-crystallized either with

the compound **4** or **8**, under the conditions containing 100–300 mM potassium sodium tartrate, 100 mM sodium citrate (pH 5.6), and 2.0–2.6 M ammonium sulfate, at 20 °C. The crystals were soaked into the cryoprotectant solution containing 60 mM potassium sodium tartrate, 25 mM sodium citrate (pH 5.6), and 2.15 M lithium sulfate and then flash frozen by plunging into liquid nitrogen. X-ray diffraction data were collected at the beamline BL45XU of the SPring-8 (Hyogo, Japan) with the aid of an automatic data-collection system ZOO[61]. Diffraction data were integrated and scaled with the programs KAMO[62] and XDS[63]. Data processing statistics are summarized in Supplementary Table S4. The structures were determined by the molecular replacement method with the program PHASER[64], using the structure of *Tte* PreQ$_1$ aptamer domain (PDB ID: 6E1S) as a search model. The solutions were refined by simulated annealing to uncouple the working and free *R* values in the refinement process by PHENIX[65]. We also performed rigid body, energy minimization, restrained isotropic B-factor, and TLS refinement with PHENIX, generating clear electron density maps corresponding to each compound. The TLS groups were identified automatically by the implemented tool in phenix.refine.The atomic models were built manually with COOT[66] and improved by iterative cycles of refinement with PHENIX. The current co-crystal structures with the compounds **4** and **8** were refined to an $R_{work}/R_{free}$ of 0.184/0.193 and 0.198/0.214 at 2.15 and 2.25 Å resolution, respectively. Atomic coordinates and the structure factors of the co-crystal structures with the compounds **4** and **8** have been deposited in the Protein Data Bank, under the accession codes 8YAM and 8YAN, respectively.

### Structure probing of different riboswitches using SHAPE-Map

**In vitro transcription for RNA preparation.** Ultramer® DNA Oligos purchased from IDT (Supplementary Table S1) were amplified using Q5® High-Fidelity DNA Polymerase (NEB) following the manufacturer's instructions and used as templates for in vitro transcription. A list of primers used for template amplification can be found in Supplementary Table S2. Using the HiScribe™ T7 High Yield RNA Synthesis Kit (NEB), RNA was transcribed following the manufacturer's guidelines and purified using RNA Clean and Concentrator−5 kit (Zymo).

**Acylation of RNA in vitro using SHAPE reagent.** 10pM of the purified RNA in 12 μL of nuclease-free water was denatured by heating at 95 °C for 2 min and snap-cooled on ice. 3.3X PreQ$_1$ folding buffer (165 mM Tris, pH 7.5, 333 mM KCl, 3.3 mM MgCl2) was used for folding the RNA by incubating at 75 °C for 5 min and slowly cooled to RT for 30 min. The folded RNA was incubated with DMSO (control), PreQ$_1$ (cognate ligand), and **4** (synthetic ligand) with a final concentration of 2.5%, 10 μM, and 150 μM respectively, for another 15 min[51]. For acylation, 5 pmol of each RNA was either treated with 1 μl DMSO (-) or was chemically modified using 100 mM 2A3 (+) by incubation at 37 °C for 20 min. The acylation was quenched using 1 M DTT and purified using Illustra G-25 columns.

**Reverse transcription of RNA for mutational profiling (MaP).** 1 μl of dNTPs (10 mM each, NEB) and 1 μl RT oligo (20 μM) (Supplementary Table S2) were added to +/− RNA followed by incubation at 70 °C for 5 min and snap cooled for a min. Next, first strand buffer mix containing 4 μl of (5X, 250 mM Tris-HCl pH 8.0, 375 mM KCl), 2 μl DTT (100 mM), 1 μl RNase Inhibitor, 1 μl SuperScript II (Invitrogen), 1 μl MnCl$_2$ (120 mM) was added, followed by incubation at 25 °C for 5 min. The +/1 samples were reverse transcribed by incubating at 42 °C for 2 h, followed by heat-inactivation of the enzyme by incubating at 75 °C for 20 min. The resulting cDNA was purified using Illustra G-25 spin columns.

**Library generation and data analysis.** The libraries were generated by the PCR amplification (Q5® High-Fidelity DNA Polymerase, NEB) of the above cDNA using two sets of primer pairs (Supplementary Table S2).

The libraries were purified using DNA clean and concentrator-5 (Zymo) 1 pM of each library was pooled for paired-end sequencing on an Illumina Miniseq instrument following manufacturer's instructions. SHAPEmapper software (https://github.com/Weeks-UNC/shapemapper2) using default parameters was used to generate SHAPE profiles for each of the RNA. Delta SHAPE analysis (https://github.com/Weeks-UNC/deltaSHAPE) was performed on the generated.map files. RNA structure prediction using SHAPE constraints was done on the structure editor (https://rna.urmc.rochester.edu/GUI/html/StructureEditor.html). Superfold was used to generate the arc plots (https://github.com/Weeks-UNC/Superfold).

### RNA Sequences and preparation for MSF analysis

The *Bsu* PreQ$_1$ sequence (34nt) (Supplementary Table S3) was flanked by 6 nucleotides of single-stranded RNA, and a sequence that is used to allow the annealing to the DNA handles for surface (3′ DNA splint in Supplementary Table S3) and bead (5′ DNA splint in Supplementary Table S3) attachment. The RNA was synthesized by IDT and tagged with 5′ biotin. The DNA handles were synthesized by Eurofins.

The RNA was annealed to the two DNA handles (10 μM) in Annealing Buffer (10 mM Tris-HCl, pH 7.4, 50 mM NaCl, 1 mM EDTA). Once annealed, the structures were mixed with an equal volume of GenTegraRNA (GenTegra) and were stored at − 20 °C until use. The annealed RNA structure (2 fmol) was then attached to 3 μl of MyOne T1 streptavidin beads (Invitrogen) in Oligo Buffer (OB) (1xPBS, 0.2% BSA, 0.1% NaAz) for 10 min before being washed and resuspended in 5 μl OB for loading in the instrument.

### MSF Instrument and measurement

Flow cell surfaces were functionalized with an oligonucleotide and then passivated with OB buffer after flow cell assembly. The RNA bound to the MyOne beads was injected into the flow cell and allowed to hybridize to the surface-bound oligonucleotide for 30 min in OB. After hybridization, the buffer was changed to the testing buffer (50 mM HEPES, pH 7.4; 100 mM KCl; 1 mM MgCl$_2$; 1% DMSO). Unbound beads were removed by washing, and any non-specifically bound beads were removed by increasing the magnetic force to - 25–30 pN. Subsequent experiments were performed at 22 °C, with the recording of bead vertical or Z-positions at 30 Hz.

MSF was performed using Depixus' prototype instrument[67]. A force ramp experiment (- 1–25 pN) was first recorded for each structure in a test buffer containing 1% DMSO. This experiment was used as a control for normalization as well as to identify structures that could be opened and closed. Next, **4** or PreQ$_1$ ligand was added at an increasing concentration in test buffer. For each condition, 50 ramp cycles were recorded. Stepped constant-force experiments were also performed. To do so, the applied magnetic force was held constant for a fixed time (- 30 sec) before increasing it stepwise. For constant-force experiments, the beads were submitted to the same force for 30–60 min.

### Analysis of ramp experiment data

We only analyzed the signal of beads whose measured variance in position was roughly equal to the noise expected from Brownian motion. Signals with excessive tracking noise or detached beads were thus excluded. For each structure, each cycle was analyzed separately, and the size and corresponding applied force of abrupt jumps in bead Z-position due to unfolding and refolding of the RNA structure were detected using an algorithm based on clustering the force-extension data with the HDBScan python module[68]. We first selected parameters to define analyzable RNA structures: (1) An unfolding force between 5–15 pN and an unfolding size between 10–25 nm in control (DMSO) conditions; (2)Total number of analyzable cycles > 20; (3) The proportion of cycles with an unfolding event > 80%. The structures were present in more than half of the conditions tested. Structures were

considered folded when analyzable cycles contained no unfolding or refolding events.

For each analyzable structure, the median forces at which unfolding and refolding occurred in DMSO conditions were used to normalize all cycles of all other conditions. The normalized unfolding and refolding forces were plotted to study the changes in distribution with different concentrations of **4**, and the median of the force at the maximum peak of the distribution for each structure was plotted against a concentration of **4**. For $PreQ_1$ ligand binding, a threshold force corresponding to the 0.95 percentile of the unfolding force distribution was determined using DMSO conditions. The fraction of force ramp cycles with an unfolding force over the threshold for each condition or did not unfold was calculated and plotted against the $PreQ_1$ ligand concentration. For both **4** and $PreQ_1$ ligand, $K_D/EC_{50}$ was calculated using the following formula: $y = (u + b*x/K_D)/(1 + x/K_D)$.

## Analysis of constant-force experiments

Data were excluded from RNA structures showing high noise (i.e., a variation in extension exceeding twice the expected length of the structure within a $1 s$ window), or structures that did not fold and unfold in response to the applied magnetic force or that did not form stable folded structures in the presence of $PreQ_1$ ligand. A simple heuristic Hidden-Markov-Model was then used to attribute a state (folded or unfolded) to each data point (the frequency of data collection was 30 Hz), with the data being presented as a trace of bead position versus time. This allowed the time the structure spent consecutively in either state before re-/unfolding (lifetime) to be assessed. The distribution of lifetimes for each individual structure and state was then analyzed, and Maximum Likelihood Estimation (MLE) was used to estimate whether one, two, or three exponential distributions were present. The Bayesian information criterion (BIC) was used to confirm that for all conditions, a single exponential distribution was the preferred model, except for the folded state lifetimes when the $PreQ_1$ ligand was present for which a mixture of two exponential distributions with two distinct lifetimes was the preferred model. To assess ligand concentration dependency, we computed the mean observed lifetime of the folded and unfolded states for the DMSO control and different concentrations of **4**. For the $PreQ_1$ ligand, we used the lifetimes obtained from the MLE analysis, together with the weights of the two distributions. To aggregate data from multiple beads, it was necessary to first compensate for minor differences in paramagnetic bead magnetism by normalizing each RNA structure with its control condition. This was achieved through analyzing the logarithm of the un-/refolding rates log(1/lifetime), which, according to the Kramers-Bell theory[69] scales linearly with the applied force.

## In vivo GFPuv reporter assay

The construct for the *Ssa*-$PreQ_1$ riboswitch-GFPuv reporter assay was designed following Dutta et al.[54]. Successful insertion of the desired RNA sequence was confirmed using Sanger sequencing. The construct was then transformed into competent *E. coli* strain JW2765 *ΔqueF* (Coli Genetic Stock Center, Yale University). Cells were streaked in the presence and absence of DMSO (control), $PreQ_1$(natural ligand), **4**, **8**, **9** (synthetic ligands), and empty vector and were grown on specialized CSB agar media at 37 °C overnight. The plates were visualized for GFP fluorescence on a UV-transilluminator at 365 nm and photographed.

## Synthetic procedures and characterization

General chemistry methods for compound synthesis and characterization are provided in the Supplementary information.

## Reporting summary

Further information on research design is available in the Nature Portfolio Reporting Summary linked to this article.

## Data availability

The data generated in this study are available in supplementary information or public repositories. Abasic mutants at positions 13, 14, and 15 (ab13_14_15) of the $PreQ_1$ riboswitch aptamer domain from *Tte* co-crystallized with compounds **4** and **8**, have been submitted to the Protein Data Bank for the details on atomic coordinates and structure factors under the accession codes 8YAM and 8YAN respectively. The sequencing reads used for SHAPE-MaP were deposited into the NLM/NCBI Sequence Read Archive (SRA) under the BioProject ID-PRJNA1077397. The source data used in Fig. 1 and Supplementary Figs. S1, S2, S3, S4 are provided as a Source Data file.

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

## Acknowledgements

The project was supported by the intramural program of the National Institutes of Health, National Cancer Institute, and Center for Cancer Research (Grant 1-ZIA-BC011585-10 to J.S.S.). For MST instrumentation and data analysis, we thank Dr. Sergey G. Tarasov and Marzena Dyba from the CCR Center for Structural Biology Biophysics Resource. We also thank Professor Joseph E. Wedekind for the generous gift of the pEnv8(GAAA) plasmid that was re-engineered in the study. For the data analysis support, this project has also been funded in part with Federal funds from the NCI, National Institutes of Health, Department of Health and Human Services, under Contract No. 75N91019D00024. The content of this publication does not necessarily reflect the views or policies of the Department of Health and Human Services, nor does mention of trade names, commercial products, or organizations imply endorsement by the U.S. Government. X-ray diffraction data were collected at BL45XU of SPring-8 (Hyogo, Japan) with the approval of the Japan Synchrotron Radiation Research Institute (JASRI) under proposal number 2021B2715. This work was supported in part by the Japan Society for the Promotion of Science (JSPS KAKENHI Grant Numbers 20K21281 and 20H02916) and a grant from The Uehara Memorial Foundation to T.N.

## Author contributions

S.P. designed experiments, performed MST measurements, fluorescence titrations, SHAPE-MaP experiments, data analysis, and in vivo translation assays, and helped write the manuscript. D.D.B., R.B., and P.R.P. performed synthetic chemistry and characterization. C.C. performed fluorescence titrations and synthetic chemistry. Z.W., H.L., K.S., A.F., and J.O. performed single molecule spectroscopic experiments and analysis. P.H. helped with bioinformatics and data analysis. T.N. designed, performed, and analyzed transcription termination assay and x-ray co-crystallography. J.S.S. conceived the project, designed experiments, and helped write the manuscript. All authors contributed to writing and editing the manuscript.

## Funding

## Competing interests

Zhen Wang, Henning Labuhn, Krishshanthi Sinnadurai, Adeline Feri, and Jimmy Ouellet are employed by Depixus SAS, who manufactures instruments used in this study. The authors declare no other competing financial or non-financial interests.
