## [Peer Review File · Nature Communications]

Mechanistic Analysis of Riboswitch Ligand Interactions Provides Insights into Pharmacological Control over Gene ExpressionREVIEWER COMMENTS

Reviewer #1 (Remarks to the Author):

In this manuscript, Parmer et al describe a very interesting series of studies aimed at (1) identifying small molecules able to bind the *Thermoanaerobacter tengcongensis* PreQ1 riboswitch, (2) using X-ray crystallography to understand how newly identified compounds bind the riboswitch at an atomic level, and (3) combining these results with single-molecule force experiments to develop mechanistic insight regarding gene regulation by the PreQ1 riboswitch. This is an important area in many respects. Ribozymes themselves are of significant interest, and the small molecule - RNA recognition field continues to undergo rapid expansion. There are also very few X-ray crystal structures of RNAs with bound small molecules. Finally, the manuscript has an innovative combination of structural data in a static context (X-ray crystallography) with data obtained in a dynamic environment (force measurements). As such, the work is important and appropriate for Nature Communications.

Despite this enthusiasm, in several instances I think the manuscript would benefit from revisions. Some of these are minor, while others will likely require additional experiments.

1) What's the KD for PreQ1 itself? This is needed for comparison with compounds 1-9

2) The authors report KDs measured using three different techniques for compounds 1-9. It is not at all apparent why some were measured by intrinsic fluorescence (1, 2, 4) while others were measured using MST (3, 5, 6, 7, 9) and one by FIA (8). Some rationale for this should be provided (is intrinsic fluorescence just faster and easier, and so that was done for 1, 2, and 4 rather than MST? The compounds analyzed by MST look like they should also be fluorescent. Why was 8 analyzed by FIA?). Given that 4 was also analyzed by MST, it would be useful to include these values in Figure 1 to show that at least for this compound the KDs correlate reasonably well between the two techniques.

3) Why were compounds 4 and 8 selected for further study? These are not the tightest binding compounds according to Figure 1; 8 is one of the worst.

4) While 8 has a weak KD (104 micromolar), its performance in a single-round transcriptional termination is quite good ($T(50) = 6.8$ micromolar, and actually better than 4. What's the reason for the discrepancy with the KD?

5) The authors do not present any in vitro (binding) specificity experiments at all. This is a significant problem, as they hypothesize later in the text that compound 8's lack of activity in cells may be due to off-target binding.

6) X-ray crystal structures are obtained using a triple-abasic mutant. Why was this mutant chosen? Do compounds 4 and 8 bind this mutant with similar KDs to the parent sequence?

7) The authors highlight stacking interactions by 4 and 8 in the crystal structure, but given that both compounds are charged at neutral pH, it also seems likely that electrostatic interactions with the sugar-phosphate backbone are important. Figure 2 E and F suggests (to me at least) that this is indeed the case. Could the authors discuss this aspect of the interactions?

8) For the force experiments, how do the authors control RNA density on the surface? Were different densities tested to ensure no site-site interactions?

9) MAGNA data in Figure 4c is fit to a binding curve, and an EC50 was calculated. Given the amount of

error in the measurement and the overall low dynamic range, this could just as easily have been fit to a straight line. Data in supplementary figure 5C suffers from similar issues (although it is nice to see that the folding/refolding forces are nearly equivalent at equivalent concentrations of compound 4). How did the authors establish that this is a statistically reasonable way to fit the data?

Reviewer #2 (Remarks to the Author):

The manuscript by Schneekloth & coworkers presents the design of chemically diverse synthetic ligands for PreQ1 riboswitches, guided by structural insights. Through multiple X-ray co-crystal structures, they confirm that these synthetic ligands share a common binding site with the natural ligand, despite their chemical diversity. Furthermore, structure probing assays reveal that one of the synthetic ligands induces conformational changes similar to those caused by the natural ligand in six different PreQ1 riboswitch aptamers. Single-molecule force spectroscopy demonstrates that the ligands stabilize the riboswitches through different mechanisms. While the natural ligand promotes the formation of a persistent, folded pseudoknot structure, a synthetic ligand slows down unfolding through a kinetic mechanism. Biochemical assays and a GFP reporter system demonstrate the functional activity of one ligand in regulating gene expression in live cells.

Overall, the study is thoroughly done, it brings together various modern methods in a breadth that is rarely seen and which I highly appreciate. A key message is that for preQ1 riboswitches (which are potential antibiotic targets), structure-guided design can create small libraries of structurally unrelated hydrophobic small molecules with individuals that can affect gene expression in living cells by altering conformational changes in RNA structures through different mechanisms. The study is certainly of great importance to the RNA field and should be given a chance to be published after proper revision of the issues listed in the following:

Lines 63-69: The differences between Class I, II, III are described in such a way that the reader is still left wondering what the differences are and what distinguishes the classes. Please improve.

Line 74: for glmS rs, citations are a bit outdated; see e.g. Stängle, D. et al. Carba-Sugar Analogs of Glucosamine-6-Phosphate: New activators for the glmS riboswitch. Chem. A Eur. J. 29, e202202378 (2023).

Figure 1: There is a difference between Figure 1 in the text and Figure 1 at the end of the manuscript. The one at the end is missing compound 9?! And compound 8 is drawn in a different orientation. Please clarify.

Line 117: Is MST a reliable method for K_d determination of the preQ1 system?! The difference in size/molecular weight/mobility is tiny for ligand-bound versus unbound RNAs. Why didn't the authors use MST for all their K_d determinations? What does the original data look like?

Line 148-152: just a comment: it is always surprising that the replacement of nucleosides by spacer riboses (without bases) for the preQ1 system leads to apparently very reliable results and important insights into ligand binding in pockets. (I also refer to the earlier publication on preQ1 binders by the same authors)

Lines 216 to 218. Please add the preQ1 class (and type in case of class I) to each organism. Otherwise it is difficult to follow.

Lines 219 to 257: I appreciate the SHAPE MaP analysis and agree that the results are convincing.

However, the authors should provide more information on the discrepancy between the secondary structure scheme for e.g. the Tte preQ1 rs shown in their Fig. 3A and the typical stem-loop and pseudo-knot interactions usually found for the same system in the literature. Fig. 3A shows a 3-way junction for Tte preQ1 rs?!

Figure 3: Increase the font size of the nucleotides in the sequence/secondary structure displays.

Lines 265 to 257: The experiments and interpretations are well described (and SupFig. !) However, I somehow miss a discussion reflecting the different Kds of compound 4 (low μM) and preQ1 (low nM) on what they authors observe in the magnetic force spectroscopy experiments. To my understanding, this huge difference is the main source for the differences they observe (e.g. in Fig. 4D).

Lines 367 to 369: Is this really the correct conclusion for compound 4? It is difficult to directly demonstrate whether compound 4 actually binds to RNA in the cell. (and the authors did not)

Reviewer #3 (Remarks to the Author):

Summary

RNA is a key player in many gene regulatory pathways and is viewed increasingly as a drug target. Understanding how small molecules interact with RNA to alter 3-D structure, dynamics and folding is essential to lay the foundation for RNA-targeting small molecules that bind with affinity and specificity to alter biological function. In this respect, the class I preQ1 riboswitch represents a model system to explore small molecule targeting of a bacterial gene-regulatory RNA. Using a structure-guided approach, Parmar and co-workers designed and synthesized nine small molecule ligands with new heterocyclic cores or sidechains to target the preQ1 riboswitch aptamer. Multiple ligands directly bind, including compound 4, which showed $\sim 16 \mu\text{M}$ affinity as well as efficacy in transcription termination assays. High-quality crystal structures reveal the mode of binding by compounds 4 and 8, which are recognized in distinctly different manners compared to preQ1. Chemical modification analysis of various preQ1 riboswitch sequences derived from diverse family members showed that compound 4 elicited conformation changes similar to the native ligand preQ1. Use of magnetic bead force tracking revealed that compound 4 influence riboswitch fold stability but has different effects compared to preQ1. PreQ1 and 4 clearly elicited gene modulation activity in live bacteria suggesting these molecules target the riboswitch to control gene regulatory conformations. This manuscript demonstrates how structure-guided compounds can be developed to target gene-regulatory RNAs. Compared to the native preQ1 ligand, the work also shows that such compounds can use different modes of RNA binding, elicit altered folding and dynamics, and change RNA stability. Such compounds can also penetrate cells and regulate genes. Overall, this work reveals several existing results based on an interdisciplinary approach that analyzes compound binding using multiple orthogonal approaches. The manuscript will be of significant interest to the journal readers. However, several comments must be addressed before further consideration.

Comments

Line 62. The modes of ligand recognition are identical in the class II and III riboswitches despite different folds, which means the description on line 62 is not entirely correct.

Lines 65-69. The authors wrote, "Class II PreQ1 riboswitches possess a distinct architecture, with the aptamer and expression platforms segregated in sequence. This class modulates gene expression

through conformational changes upon PreQ1 binding.^{22, 23} Notably, Class III riboswitches are characterized by a pseudoknot structure formed by the aptamer domain, enabling ligand recognition and gene regulation.²⁴ The description is actually the opposite of what is written for class II and III riboswitches. Class II riboswitches form a pseudoknot at the floor of the preQ1 pocket that directly connects ligand binding to gene regulation, since the pseudoknot buries the Shine-Dalgarno sequence. By contrast, class III riboswitches use separate aptamer and expression platform domains. The class III riboswitch separates the expression platform from the ligand pocket by >40 Å. The reviewer recommends altering the text (i.e., class II becomes III and III becomes II).

Line 80. It is best to state here that the class I preQ1 riboswitch is being targeted.

Line 159. "strict hydrogen bonds". It is better to say "strictly conserved hydrogen bonds".

Lines 177-178. Differences in the L2 loop and S2 are interesting for compound 4 and 8. However, the authors noted that some differences could be due to abasic sites used at 13,14,15. A recent paper on the Tte riboswitch (Schroeder et al 2020 NAR) described differences in L2 and S2 in the apo state compared to the bound state. Unlike previous Tte structures in which L2 was disordered, L2 is intact in the recent structures (PDB codes 6vuh apo & 6vui preQ1 bound). Do the structures in complex with compound 4 or 8 look more like apo or bound states? The nucleotides in loop L2 move from an unstacked conformation (apo) to a stacked conformation (bound), which is argued to support gene regulation. It would be interesting to know if 4 or 8 supports an intermediate conformation as well.

Line 192. Please provide a manuscript reference for PDB ID 6e1v.

Line 203. Please provide a manuscript reference for PDB entry 3q50.

Line 210. Insert reference 25 here (McCown et al 2014 Chem Biol 21, 880-889) to support the statements about preQ1 riboswitches.

Line 215. The authors should clarify that the sequences examined have representatives from all three preQ1 riboswitch classes.

Lines 229-232. The text reads "Specifically, the Bsu riboswitch showed stabilization of structure at C20, A21, C22 belonging to aptamer domain, (consistent with crystallographic studies),¹⁸ as well as at the C53 and U60 bases within the terminator domain". Reference 18 here does not correspond to the Bsu crystal structure, which is ref. 20. Also, should the terminator region analysis be interpreted in light of the Bsu NMR analysis? Kang et al. (2009) Mol Cell. 33, 784

Lines 219-228. The use of SHAPE-map provides important information about the ability of the ligand to modulate the RNA structure. However, the average reader will not understand the significance of the chosen sequences, which appear to belong to the three known preQ1 riboswitch classes. This point and the rationale must be provided; the figure 3 shows the different class designation and this should be noted in the legend. A particularly difficult aspect of the analysis to understand is that the predicted secondary structures do not conform to the consensus models of these riboswitches described in ref. 25. Nor do the models appear to integrate the known long-range interactions from crystallography or NMR. As the authors know, folding riboswitches can be difficult due to the absence of the free-energy contribution of the ligand and the difficulty that most programs have with pseudoknots. As it stands, most of the folds in Figure 3 are inaccurate depictions. Are the authors actually trying to convince the reader that these are relevant gene-regulatory states influenced by ligands? For example, the Lrh riboswitch with preQ1 appears to cause very low SHAPE reactivity at the specificity base C28, but the ribosome binding site is much more reactive. By contrast, compound 4 produces a much more reactive

specificity base but the ribosome binding site is much less reactive. A better approach would be to map the SHAPE reactivity onto the known secondary structures and tertiary contacts derived from thermodynamic equilibrium states determined NMR/crystallography. The regions could then be compared using the existing color code. The location of ligand binding should be indicated. In its current form, there is a disconnect between the predicted structures, gene regulation and ligand binding.

Another point worth considering for the SHAPE-Map studies is that transcriptional riboswitches fold under kinetic control. Therefore, the addition of the terminator hairpin to the transcribed Bsu product will result in a conformation that is more stable than the antiterminated pseudoknot that binds ligand (see Kang et al. (2009) Mol Cell. 33, 784). Indeed, few differences are seen for the Bsu sample even though A36 and U12 should bind preQ1 but not compound 4. Perhaps the lack of modulation at the latter two positions supports the more stable terminated conformation shown?

Lines 253-255. The reviewer urges the authors to consider a more useful chemical modification mapping approach. The crystal structures are known, so why not use them? The passage, "While these secondary structures are experimentally informed, they do not necessarily reflect three-dimensional aptamer structure with perfect accuracy" is an understatement. Again, the reviewer questions the value of interpreting the data using 2-D structures that are folded incorrectly.

Figure 2. Please state that the ligands were omitted from the phase calculation for the maps shown.

Figure 4c and Supp Fig 5c. Please use consistent units and significant figures. The left panel of Fig 4c uses 1/100th of a μM for measurement but 1/10th for errors. The right uses 1/100th of a nM. The text on line 308 reports 1 μM and 1 nM.

Figure 4e appears to plot the same analysis as Suppl. Fig. 5i. The refolding rate (1/unfolded lifetime) of the PreQ1 riboswitch in constant force experiments with increasing concentration of 4. However, the data show opposite trends $>100 \mu\text{M}$ compound 4. Please clarify what each of the plots is showing and how the data were obtained.

Supplemental. Isn't it more accurate to label the section on Fluorescence Intensity as Fluorescence Anisotropy since this is the basis of the fluorescence changes, which will be more familiar to the reader?

Supplemental. For the structure determination, the same crystal form appears to have been described previously, which was used for MR. Please state whether the test set was preserved from this previous structure to avoid overfitting of the new structures as recommended [Kleywegt & Brünger (1996) Structure 4, 897]. The authors used TLS parameters in their previous search model. Was TLS refinement conducted here? If so, please state how the TLS groups were chosen. For consideration, the report of Rfree values in the Supplemental text is incomplete without providing the corresponding Rwork values, which should be within 6% to indicate that overfitting did not occur. For example, a reported Rfree of 0.193 with an Rwork of 0.130 would be problematic.

Supplementary Fig 1. Many of the curves do not show binding saturation, making it difficult to fit the KD. This resulted in large errors. The reviewer recommends adding additional concentration points at 1 mM ligand levels. Alternatively, the authors should qualify for the reader that these values are only estimates of affinity due to the lack of saturation. This is acceptable only because these compounds are not the main focus of the manuscript. The figure should also use consistent significant figures.

Supplementary Fig 3. Why is there a doublet for the read-through product? This should be noted in the

approach.

Supplementary figure 4. What does “delta” mean? Is it the reactivity of the apo state minus the bound state or vice versa? For clarity please add the riboswitch sequence and secondary structure from the folded state above the delta shape reactivity profiles. This will allow the reader to follow the arguments in the text about regions that change. A particularly difficult aspect about the interpretation of this result is that the reader does not know what regions of the riboswitches are modulating because they have not been provided. To be effective, the reader should have the corresponding secondary structures and their corresponding structural elements must be labeled (e.g., P1, P2, etc, L1, L2 etc). Also, the corresponding expression platforms must be labeled. Please state the significance of the green versus purple highlighting.

Supplementary table 4. The reported $I/s(I)$ is different from the PDB validation reports. Table S4 reports compound 8 as 1.44 but the PDB states 1.43; Table S4 reports compound 4 as 1.57 but the PDB states 1.23. Please correct this error. Also, the Greek letter sigma has been replaced by S from the Latin alphabet. Please use Σ in the footnotes to indicate summation. Also, the PDB Validation Reports provided are not the versions required for manuscript review, meaning the coordinates have not been validated by the PDB. Please provide valid reports.

Reference 51 is duplicated from 23. Please check the other references for duplications.

RESPONSES TO REVIEWER COMMENTS

Reviewer #1 (Remarks to the Author):

In this manuscript, Parmer et al describe a very interesting series of studies aimed at (1) identifying small molecules able to bind the *Thermoanaerobacter tengcongensis* PreQ₁ riboswitch, (2) using X-ray crystallography to understand how newly identified compounds bind the riboswitch at an atomic level, and (3) combining these results with single-molecule force experiments to develop mechanistic insight regarding gene regulation by the PreQ₁ riboswitch. This is an important area in many respects. Ribozymes themselves are of significant interest, and the small molecule - RNA recognition field continues to undergo rapid expansion. There are also very few X-ray crystal structures of RNAs with bound small molecules. Finally, the manuscript has an innovative combination of structural data in a static context (X-ray crystallography) with data obtained in a dynamic environment (force measurements). As such, the work is important and appropriate for Nature Communications.

Thank you for the helpful and constructive review of our manuscript.

Despite this enthusiasm, in several instances I think the manuscript would benefit from revisions. Some of these are minor, while others will likely require additional experiments.

1) What's the KD for PreQ₁ itself? This is needed for comparison with compounds 1-9

The K_D of natural ligand PreQ₁ with *Tte*-PreQ₁ riboswitch as determined previously is 11.7 ± 0.35 nM by MST (ACS Cent. Sci. 2022, 8, 6, 741–748) and with *Bsu*-PreQ₁ riboswitch is 4.1 ± 0.6 nM by FIA (Nature Communications, 2019, volume 10, Article number: 1501). We agree with the reviewer that this is important to include, and it is now in the revised manuscript.

2) The authors report KDs measured using three different techniques for compounds 1-9. It is not at all apparent why some were measured by intrinsic fluorescence (1, 2, 4) while others were measured using MST (3, 5, 6, 7, 9) and one by FIA (8). Some rationale for this should be provided (is intrinsic fluorescence just faster and easier, and so that was done for 1, 2, and 4 rather than MST? The compounds analyzed by MST look like they should also be fluorescent. Why was 8 analyzed by FIA?). Given that 4 was also analyzed by MST, it would be useful to include these values in Figure 1 to show that at least for this compound the KDs correlate reasonably well between the two techniques.

Thank you for the opportunity to better communicate the reasons behind reporting results this way. The affinity analysis was initially conducted using an MST instrument with

cy5-labeled RNA for **all** the compounds. However, compounds **1,2,4** showed fluctuations in fluorescence during the initial capillary-scan illustrating that with the change in the concentration of ligand, there was initial fluorescence quenching. In these situations, it is not appropriate to perform MST since variations in initial fluorescence can cause fluctuations in thermophoretic efficiency (this is described by the manufacturer here <https://nanotempertech.com/nanopedia/ligand-induced-fluorescence-change>). Therefore, this fluctuation in initial fluorescence was used for the determination of K_D by plotting it as a function of RNA concentration. By contrast, the other compounds showed no change in initial fluorescence but did exhibit a dose-dependent effect on thermophoresis, which could be used to measure K_D . The exception to this was that compound **8** had no apparent change in fluorescence or thermophoresis and therefore an inherent ligand fluorescence intensity assay using a more sensitive fluorimeter was used for affinity measurement. The reasoning for this is explained in the manuscript (Page 4, lines ~120-140). In addition, Figure S1(A) has been redrawn in the supplementary information by normalizing the data for clear representation.

3) Why were compounds 4 and 8 selected for further study? These are not the tightest binding compounds according to Figure 1; 8 is one of the worst.

We apologize for the lack of clarity here. **All** the compounds synthesized in the study were subjected to transcription termination assay to check for biochemical activity. Out of all the compounds, only compounds **4** and **8** showed improved activity in *in vitro* single-round transcription termination assays. Other compounds displayed poor solubility or aggregation issues when higher concentrations were required to perform the assay in aqueous buffers (page 7, lines 166-170). Despite a relatively weak K_D of **8**, it showed improved biochemical activity and therefore was used for further analysis. Thus, only **4** and **8** had both good binding, solubility, and improved activity in functional assays.

4) While 8 has a weak K_D (104 micromolar), its performance in a single-round transcriptional termination is quite good (T_{50}) = 6.8 micromolar, and actually better than 4. What's the reason for the discrepancy with the K_D ?

The reasons for this are not completely clear. However, in our hands it is not uncommon to see a disconnect between ligand affinity and activity for riboswitch ligands (for example, see Connelly, et al., *Nature Communications* **2019**, 10:1501 and Abulwerdi, et al., *J. Med. Chem.* **2016** 22;59(24):11148-11160). While we often attribute this observation to differences in mechanism of recognition/binding, it is typically extremely difficult to rationalize without extensive mechanistic studies. Since we did see consistency in binding and activity with **4**, we elected to invest resources in focusing on its mechanism as described in the paper. In response to the reviewer's comment, we have added a sentence commenting on this in the revised manuscript on lines 166-170.

5) The authors do not present any in vitro (binding) specificity experiments at all. This is a significant problem, as they hypothesize later in the text that compound 8's lack of activity in cells may be due to off-target binding.

We agree with the reviewer that this is an important set of data to include. In the revised manuscript, we have now performed binding assays with six different RNAs varying in sequence and structure, including RNA G-quadruplexes from the DHX15 and the Mango aptamer, a DNA G-quadruplex from the MYCN gene, the SARS CoV-2 frameshifting pseudoknot, one small RNA stem loop, and the snRNA U4/U6 3-way junction. Figure S3 in the supplementary information shows affinity measurements performed with different structures to present evidence of specificity. Here, we observe no binding to any of these structures for **4**, indicating that it does not bind non-specifically to nucleic acid structures. This has also been noted in the revised manuscript on lines ~135-139. While this does not provide evidence for RNA off target binding, we argue that it is important to note that structurally similar compounds do have promiscuous nucleic acid binding behavior, as mentioned.

6) X-ray crystal structures are obtained using a triple-abasic mutant. Why was this mutant chosen? Do compounds **4** and **8** bind this mutant with similar KDs to the parent sequence?

We used a triple-abasic mutant in the crystallography because as reported previously (Connelly et al., Nat. Commun. 2019), when the wild-type Tte-PreQ₁ riboswitch was used, we only obtained the structures indistinguishable from the ligand-free form of the RNA, in which the nucleobase of A14 occupies the PreQ₁-binding site. This result may be due to the lower affinity for the compound compared to that of the cognate ligand PreQ₁. Since the K_D and T₅₀ values of compounds **4** and **8** were larger than those for PreQ₁, we hypothesized that like the previous work, the crystallography would yield structures indistinguishable from the ligand-free form. Therefore, we used a triple-abasic mutant to obtain the complex structures. However, we believe that the co-crystal structures of the triple-abasic mutant with the compounds exhibit biologically relevant forms based on the present results of comprehensive biochemical analyses and homology to other reported structures.

7) The authors highlight stacking interactions by **4** and **8** in the crystal structure, but given that both compounds are charged at neutral pH, it also seems likely that electrostatic interactions with the sugar-phosphate backbone are important. Figure 2 E and F suggests (to me at least) that this is indeed the case. Could the authors discuss this aspect of the interactions?

As suggested by the reviewer, we also expected hydrogen bonding interactions through heteroatoms of these compounds. However, there was only one weak hydrogen bond (3.6 Å) between compound **4** and A29 of the riboswitch as described in the submitted

version of the manuscript. In this revision, the interaction of the PreQ₁ riboswitch with these compounds was explored again, by extending the interatomic distance up to 4 Å. Consequently, we observed a weak interaction between the carbonyl group of the compound **4** and 2' OH group of G11 of the riboswitch (interatomic distance is 3.75 Å). The 2' OH group of G11 also seems to form weak interaction with the amine of the pyrrole ring of the compound **8** (interatomic distance is 3.66 Å). We added these interactions in the revised Figure 2E and F. We have also included the description of these interactions in the revised main text. We thank the reviewer for the helpful suggestion, which strengthens our structural discussion and analysis.

8) For the force experiments, how do the authors control RNA density on the surface? Were different densities tested to ensure no site-site interactions?

We thank the reviewer for raising this concern. In the prototype MAGNA instrument used, the RNA density is controlled via visual inspection. Only individual magnetic beads were tracked; any clumped or close-proximity beads were excluded. Moreover, if interference between two close beads were to occur this would cause high levels of noise in their tracking signals so they would be consequently excluded from the analysis during data cleaning. If there were multiple RNA molecules attached to the same bead, the unfolding force would be too high for the bead to show any unfolding signature to be quantified as a suitable structure for downstream analysis. Therefore, only single molecules are selected for analysis. A sentence noting this has been added to the revised manuscript.

9) MAGNA data in Figure 4c is fit to a binding curve, and an EC₅₀ was calculated. Given the amount of error in the measurement and the overall low dynamic range, this could just as easily have been fit to a straight line. Data in supplementary figure 5C suffers from similar issues (although it is nice to see that the folding/refolding forces are nearly equivalent at equivalent concentrations of compound **4**). How did the authors establish that this is a statistically reasonable way to fit the data?

We thank the reviewer for asking this question. Since we know that compound **4** binds to the RNA structure, and we assume it is a 1:1 stoichiometry similar to PreQ₁ ligand, consistent with structural and biophysical data also presented here in the manuscript, we therefore fitted the same binding equation as for PreQ₁ (Hill's equation with a scaler and offset). The fitting confidence interval is provided together with the EC₅₀ value. These assumptions are listed in the experimental section.

Reviewer #2 (Remarks to the Author):

The manuscript by Schneekloth & coworkers presents the design of chemically diverse synthetic ligands for PreQ1 riboswitches, guided by structural insights. Through multiple X-ray co-crystal structures, they confirm that these synthetic ligands share a

common binding site with the natural ligand, despite their chemical diversity. Furthermore, structure probing assays reveal that one of the synthetic ligands induces conformational changes similar to those caused by the natural ligand in six different PreQ₁ riboswitch aptamers. Single-molecule force spectroscopy demonstrates that the ligands stabilize the riboswitches through different mechanisms. While the natural ligand promotes the formation of a persistent, folded pseudoknot structure, a synthetic ligand slows down unfolding through a kinetic mechanism. Biochemical assays and a GFP reporter system demonstrate the functional activity of one ligand in regulating gene expression in live cells.

Overall, the study is thoroughly done, it brings together various modern methods in a breadth that is rarely seen and which I highly appreciate. A key message is that for PreQ₁ riboswitches (which are potential antibiotic targets), structure-guided design can create small libraries of structurally unrelated hydrophobic small molecules with individuals that can affect gene expression in living cells by altering conformational changes in RNA structures through different mechanisms. The study is certainly of great importance to the RNA field and should be given a chance to be published after proper revision of the issues listed in the following:

Thank you for the constructive comments and review of our manuscript.

Lines 63-69: The differences between Class I, II, III are described in such a way that the reader is still left wondering what the differences are and what distinguishes the classes. Please improve.

We agree that this could be improved. In the revised manuscript we have made changes in the text to improve clarity and distinction between the three classes.

Line 74: for glmS rs, citations are a bit outdated; see e.g. Stängle, D. et al. Carba-Sugar Analogs of Glucosamine-6-Phosphate: New activators for the glmS riboswitch. Chem. A Eur. J. 29, e202202378 (2023).

Thank you for this suggestion and we apologize for using outdated references. Our revised manuscript contains the more recent citations as well and we agree that these are important and relevant to include.

Figure 1: There is a difference between Figure 1 in the text and Figure 1 at the end of the manuscript. The one at the end is missing compound 9?! And compound 8 is drawn in a different orientation. Please clarify.

We apologize for this oversight. In the revised manuscript, we have re-drawn compound **8** in a more consistent orientation. Additionally, Figure 1 is now corrected and is consistent throughout to include compound **9**.

Line 117: Is MST a reliable method for K_D determination of the preQ1 system?! The difference in size/molecular weight/mobility is tiny for ligand-bound versus unbound RNAs. Why didn't the authors use MST for all their K_D determinations? What does the original data look like?

Thank you for this comment, which was noted by Reviewer 1 as well (see above for comments as well). In short, with this particular set of compounds we needed to use both fluorescence titrations and MST to measure accurate K_D values due to spectroscopic interference and alterations in initial fluorescence. We were unable to find a medium throughput biophysical assay consistent enough to be used for all compounds. This is now described more clearly in the revised manuscript.

From our response to Reviewer 1 above: “The affinity analysis was initially conducted using an MST instrument with cy5-labeled RNA for **all** the compounds. However, compounds **1,2,4** showed fluctuations in fluorescence during the initial capillary-scan illustrating that with the change in the concentration of ligand, there was initial fluorescence quenching. In these situations it is not appropriate to perform MST since variations in initial fluorescence can cause fluctuations in thermophoretic efficiency (this is described by the manufacturer here ()). Therefore, this fluctuation in initial fluorescence was used for the determination of K_D by plotting it as a function of RNA concentration. However, for **8** no apparent change in fluorescence or thermophoresis was observed and therefore, fluorescence intensity assay using a more sensitive fluorimeter was used for affinity measurement.”

Line 148-152: just a comment: it is always surprising that the replacement of nucleosides by spacer riboses (without bases) for the preQ1 system leads to apparently very reliable results and important insights into ligand binding in pockets. (I also refer to the earlier publication on preQ1 binders by the same authors)

Thank you for noticing this – we also find it an intriguing observation. Since Reviewer #1 also noted this, we felt it was noteworthy to add context to the revised manuscript as well. As described in the response to Reviewer #1: We used a triple-abasic mutant in the crystallography because as reported previously (Connelly et al., Nat. Commun. 2019), when the wild-type Tte-PreQ₁ riboswitch was used, we only obtained the structures indistinguishable from the ligand-free form of the RNA, in which the nucleobase of A14 occupies the PreQ1-binding site. This result may be due to the lower affinity for the compound compared to that of the cognate ligand PreQ₁. Since the K_D and T_{50} values of compounds 4 and 8 were larger than those for PreQ₁, we hypothesized that like the previous work, the crystallography would yield structures indistinguishable from the ligand-free form.

Therefore, we used a triple-abasic mutant to obtain the complex structures. However, we believe that the co-crystal structures of the triple-abasic mutant with the compounds exhibit biologically relevant forms based on the present results of comprehensive biochemical analyses and homology to other reported structures.

Lines 216 to 218. Please add the PreQ₁ class (and type in case or class I) to each organism. Otherwise, it is difficult to follow.

We agree that more explicit information about riboswitch class would help with clarity and have added this information to the revised manuscript.

Lines 219 to 257: I appreciate the SHAPE MaP analysis and agree that the results are convincing. However, the authors should provide more information on the discrepancy between the secondary structure scheme for e.g. the Tte preQ₁ rs shown in their Fig. 3A and the typical stem-loop and pseudo-knot interactions usually found for the same system in the literature. Fig. 3A shows a 3-way junction for Tte preQ₁ rs?!

We apologize for this inconsistency. In the revised manuscript, we have now re-drawn the secondary structure diagrams to more accurately reflect the pseudoknot structures that these riboswitches adopt in the presence of ligands. Thank you for this comment and we believe the revised figure both more accurately represents the complex structures of the aptamers and provides more information about their structure in response to ligand binding.

Figure 3: Increase the font size of the nucleotides in the sequence/secondary structure displays.

Thank you for this suggestion. In the revised manuscript we have increased the nucleotide font size in Figure 3.

Lines 265 to 257: The experiments and interpretations are well described (and Sup Fig. 1) However, I somehow miss a discussion reflecting the different K_Ds of compound 4 (low μM) and preQ₁ (low nM) on what they authors observe in the magnetic force spectroscopy experiments. To my understanding, this huge difference is the main source for the differences they observe (e.g. in Fig. 4D).

We thank the reviewer for raising this question. There is indeed a difference in the binding affinity between the natural ligand of PreQ₁ and compound 4, with compound 4 requiring to be at a much higher concentration to have an effect. However, we do not think the difference we observed in the mechanisms by which the two ligands stabilize the RNA structure is due to the concentration range we have tested or a consequence of the different K_D values.

We do not believe that compound **4** induces persistent, stable folded states at even higher concentrations. As an example, we show in the figure below (and attached as a PDF) (A) the traces (RNA extension vs. time) of the same RNA molecule in the control condition (top, blue), with 500 μM **4** (middle, red), and 500 nm of PreQ₁ ligand (bottom, green). We see by eye that neither control nor **4** cause the RNA to form stable folded states; PreQ₁ however does (see insets).

In (B) we show a different PreQ₁ structure, this time with control (top, blue), and 1 nm i.e. the lowest probed PreQ₁ concentration (middle, red) and 5 nm (bottom, green) of PreQ₁ ligand. We see that already for the lowest concentrations of PreQ₁ ligand, we occasionally see stable folded states (at 1 nm PreQ₁ concentration, we expect in the order of one stable state event per 30 minutes on average).

(A)

(B)

Similarly, when doing a quantitative analysis of observed lifetimes of the folded states (as presented in the manuscript), we find two distinct lifetimes in the presence of PreQ₁ ligand, whereas we find only one short lifetime in the presence of **4**. Contrary to PreQ₁ ligand, **4** does increase the lifetime of the short, folded states however, as described in the manuscript.

Therefore, we strongly believe we are observing a difference in binding mechanisms rather than purely a concentration effect. Because this is an important point, we have added these data to the supplementary information (Figure S7 in the revised SI document) and a sentence commenting on this in the manuscript. We argue the inclusion of these data strengthens the point that this is truly a mechanistic distinction rather than a concentration effect.

Lines 367 to 369: Is this really the correct conclusion for compound 4? It is difficult to directly demonstrate whether compound 4 actually binds to RNA in the cell. (and the authors did not)

We thank the reviewer for raising this concern. Since we could not observe the formation of the pseudoknot, we cannot directly compare the binding effects of PreQ₁ and compound 4. However, in our constant force experiments, we have demonstrated that compound 4 changes the kinetics of folding and refolding. Furthermore, in the ramp experiment, we have also observed an increase in the unfolding and refolding forces of the RNA structure. In our system, if the compound purely binds but does not induce any effect on the RNA's structural stability, we do not observe any differences. Therefore, the fact that we can observe any effect means that compound 4 must interact with the RNA molecule and increase its structural stability.

Additionally, while our biophysical, biochemical, and cell-based reporter gene studies are entirely consistent with RNA binding as a mechanism in cells, the reviewer is correct that we cannot completely rule out an unanticipated mechanism in cells. We have re-worded the sentence on page 19 as follows: "Finally, cells treated with 8 also did not respond, even though 8 both binds to and modulates the function of the riboswitch in vitro. Importantly, these results demonstrate that both PreQ₁ and 4 **most likely** exhibit gene modulation activity by directly binding to RNA structures in cells, rather than nonspecific or other off-target mechanisms.

Reviewer #3 (Remarks to the Author):

RNA is a key player in many gene regulatory pathways and is viewed increasingly as a drug target. Understanding how small molecules interact with RNA to alter 3-D structure, dynamics and folding is essential to lay the foundation for RNA-targeting small molecules that bind with affinity and specificity to alter biological function. In this respect, the class I PreQ₁ riboswitch represents a model system to explore small molecule targeting of a bacterial gene-regulatory RNA. Using a structure-guided approach, Parmar and co-workers designed and synthesized nine small molecule ligands with new heterocyclic cores or sidechains to target the preQ1 riboswitch aptamer. Multiple ligands directly bind, including compound 4, which showed ~16 μM affinity as well as efficacy in transcription termination assays. High-quality crystal structures reveal the mode of binding by compounds 4 and 8, which are recognized in distinctly different manners compared to PreQ₁. Chemical modification analysis of various PreQ₁ riboswitch sequences derived from diverse family members showed that compound 4 elicited conformation changes similar to the native ligand PreQ₁. Use of magnetic bead force tracking revealed that compound 4 influence riboswitch fold stability but has different effects compared to PreQ₁. PreQ₁ and 4 clearly elicited gene modulation activity in live bacteria suggesting these molecules target the riboswitch to control gene regulatory conformations. This manuscript demonstrates how

structure-guided compounds can be developed to target gene-regulatory RNAs. Compared to the native PreQ₁ ligand, the work also shows that such compounds can use different modes of RNA binding, elicit altered folding and dynamics, and change RNA stability. Such compounds can also penetrate cells and regulate genes. Overall, this work reveals several existing results based on an interdisciplinary approach that analyzes compound binding using multiple orthogonal approaches. The manuscript will be of significant interest to the journal readers. However, several comments must be addressed before further consideration.

Thank you for the thoughtful and constructive review of our work.

Comments

Line 62. The modes of ligand recognition are identical in the class II and III riboswitches despite different folds, which means the description on line 62 is not entirely correct.

We apologize for this inconsistency. In the revised manuscript we have clarified the language on line 62 to more accurately reflect the modes of ligand recognition in class II and III riboswitches.

Lines 65-69. The authors wrote, “Class II PreQ₁ riboswitches possess a distinct architecture, with the aptamer and expression platforms segregated in sequence. This class modulates gene expression through conformational changes upon PreQ₁ binding.^{22, 23} Notably, Class III riboswitches are characterized by a pseudoknot structure formed by the aptamer domain, enabling ligand recognition and gene regulation.²⁴” The description is actually the opposite of what is written for class II and III riboswitches. Class II riboswitches form a pseudoknot at the floor of the preQ₁ pocket that directly connects ligand binding to gene regulation, since the pseudoknot buries the Shine-Dalgarno sequence. By contrast, class III riboswitches use separate aptamer and expression platform domains. The class III riboswitch separates the expression platform from the ligand pocket by >40 Å. The reviewer recommends altering the text (i.e., class II becomes III and III becomes II).

We apologize for this error and thank the reviewer for pointing it out. We agree that the reviewer’s comment is correct and have revised the language in our revised manuscript to correctly refer to Class II and II PreQ₁ riboswitches.

Line 80. It is best to state here that the class I PreQ₁ riboswitch is being targeted.

We agree. Our revised manuscript includes this statement.

Line 159. “strict hydrogen bonds”. It is better to say “strictly conserved hydrogen bonds”.

We thank the reviewer for this comment and agree. We have included this language in the revised manuscript.

Lines 177-178. Differences in the L2 loop and S2 are interesting for compound 4 and 8. However, the authors noted that some differences could be due to abasic sites used at 13,14,15. A recent paper on the Tte riboswitch (Schroeder et al 2020 NAR) described differences in L2 and S2 in the apo state compared to the bound state. Unlike previous Tte structures in which L2 was disordered, L2 is intact in the recent structures (PDB codes 6vuh apo & 6vui preQ1 bound). Do the structures in complex with compound 4 or 8 look more like apo or bound states? The nucleotides in loop L2 move from an unstacked conformation (apo) to a stacked conformation (bound), which is argued to support gene regulation. It would be interesting to know if 4 or 8 supports an intermediate conformation as well.

Thank you for this helpful comment/question. According to your suggestion, we compared the structures and found that L2 of the compound-bound forms adopt an intermediate-like conformation between the PreQ₁-free and PreQ₁-bound states. We have revised manuscript to add some descriptions and prepared a new figure on this matter, which strengthens our structural discussion and analysis. We have also added the helpful reference that the reviewer noted (Reference 41 in the revised manuscript).

Line 192. Please provide a manuscript reference for PDB ID 6e1v.

Thank you for catching this omission and we apologize – the reference has now been included in the revised manuscript.

Line 203. Please provide a manuscript reference for PDB entry 3q50.

We have added the reference for this PDB entry.

Line 210. Insert reference 25 here (McCown et al 2014 Chem Biol 21, 880-889) to support the statements about preQ₁ riboswitches.

We agree with the reviewer that it is important to cite it in line 210 as well as earlier. We have added it to revised manuscript.

Line 215. The authors should clarify that the sequences examined have representatives from all three PreQ₁ riboswitch classes.

We agree that this is an important point to make. The sentence on line 216 in the revised manuscript now makes this point.

Lines 229-232. The text reads “Specifically, the Bsu riboswitch showed stabilization of structure at C20, A21, C22 belonging to aptamer domain, (consistent with crystallographic studies),¹⁸ as well as at the C53 and U60 bases within the terminator domain”. Reference 18 here does not correspond to the Bsu crystal structure, which is ref. 20. Also, should the terminator region analysis be interpreted in light of the Bsu NMR analysis? Kang et al. (2009) Mol Cell. 33, 784

Thank you for pointing this out. In the revised manuscript, reference 18 has been changed to reference 20 which informs the crystal structure of *Bsu* PreQ₁. The terminator region analysis referring to Kang et al. (2009) Mol Cell. 33, 784, is also a very useful insight and has been described into the revised text to further elaborate the binding of the ligands. The reference has been added to the text as well.

Lines 219-228. The use of SHAPE-map provides important information about the ability of the ligand to modulate the RNA structure. However, the average reader will not understand the significance of the chosen sequences, which appear to belong to the three known preQ₁ riboswitch classes. This point and the rationale must be provided; the figure 3 shows the different class designation and this should be noted in the legend. A particularly difficult aspect of the analysis to understand is that the predicted secondary structures do not conform to the consensus models of these riboswitches described in ref. 25. Nor do the models appear to integrate the known long-range interactions from crystallography or NMR. As the authors know, folding riboswitches can be difficult due to the absence of the free-energy contribution of the ligand and the difficulty that most programs have with pseudoknots. As it stands, most of the folds in Figure 3 are inaccurate depictions. Are the authors actually trying to convince the reader that these are relevant gene-regulatory states influenced by ligands? For example, the Lrh riboswitch with PreQ₁ appears to cause very low SHAPE reactivity at the specificity base C28, but the ribosome binding site is much more reactive. By contrast, compound 4 produces a much more reactive specificity base but the ribosome binding site is much less reactive. A better approach would be to map the SHAPE reactivity onto the known secondary structures and tertiary contacts derived from thermodynamic equilibrium states determined NMR/crystallography. The regions could then be compared using the existing color code. The location of ligand binding should be indicated. In its current form, there is a disconnect between the predicted structures, gene regulation and ligand binding.

Thank you for this helpful comment – we agree that the original depiction of the SHAPE-MaP data was insufficient. In response, we have significantly revised Figure 4. Several changes have been made:

1. The class designation has been noted in the figure legend as advised.

2. We have re-drawn the SHAPE-informed secondary structures by imposing SHAPE restraints on the experimentally validated secondary structures (x-ray crystallography/NMR) for much better understanding of the interactions of the ligands with the riboswitches. This more accurately reflects the true and relevant structures of the RNAs.

3. We have now also illustrated long-range interactions in the revised Figure as indicated by the reviewer. This will help build a connection between predicted structures and mode of recognition by the ligands as correctly touched upon by the reviewer.

As the reviewer suggested, we have struggled with the best way to represent these data. We now feel that the revised Figure 4 is a dramatically improved and far more accurate representation of the SHAPE data and the RNA structures. Thank you for this extremely helpful comment.

Another point worth considering for the SHAPE-Map studies is that transcriptional riboswitches fold under kinetic control. Therefore, the addition of the terminator hairpin to the transcribed Bsu product will result in a conformation that is more stable than the antiterminated pseudoknot that binds ligand (see Kang et al. (2009) Mol Cell. 33, 784). Indeed, few differences are seen for the Bsu sample even though A36 and U12 should bind PreQ₁ but not compound 4. Perhaps the lack of modulation at the latter two positions supports the more stable terminated conformation shown?

Thank you for this helpful comment. We agree that the addition of the terminator hairpin could potentially impact the conformation as the reviewer suggested. This strengthens our analysis of the data, and we have added a comment mentioning this as well as the reference to the section describing SHAPE MaP results. Interestingly, our SHAPE-MaP predicted basepairing perfectly matches the NMR structure in the reference, further highlighting the importance and relevance of that work to our own.

Lines 253-255. The reviewer urges the authors to consider a more useful chemical modification mapping approach. The crystal structures are known, so why not use them? The passage, “While these secondary structures are experimentally informed, they do not necessarily reflect three-dimensional aptamer structure with perfect accuracy” is an understatement. Again, the reviewer questions the value of interpreting the data using 2-D structures that are folded incorrectly.

Thank you for the comment. In response to this reviewer’s comments, and as described above, we have remade the secondary structures using the information from X-ray structures and NMR data, and the pseudoknots now are in accordance with the literature

and previous reports which make the analysis much clear for the readers to understand (and consistent with other experimental data).

Figure 2. Please state that the ligands were omitted from the phase calculation for the maps shown.

We have added this statement to the revised version.

Figure 4c and Supp Fig 5c. Please use consistent units and significant figures. The left panel of Fig 4c uses 1/100th of a μM for measurement but 1/10th for errors. The right uses 1/100th of a nM. The text on line 308 reports 1 μM and 1 nM.

Thank you for this comment. We have corrected the inconsistency with significant figures/error reporting as requested. However, due to the large differences between the binding affinity (nM vs μM), it is challenging to clearly illustrate these data the both in μM or nM, therefore we chose to show the concentration in different units. If data are illustrated all in the same unit, the curves become very difficult to interpret. We note that units are clearly labeled multiple times in Figure 5 C, E, and F as well as Figure S5c.

Figure 4e appears to plot the same analysis as Suppl. Fig. 5i. The refolding rate (1/unfolded lifetime) of the PreQ1 riboswitch in constant force experiments with increasing concentration of 4. However, the data show opposite trends $>100 \mu\text{M}$ compound 4. Please clarify what each of the plots is showing and how the data were obtained.

We thank the reviewer for raising this question. These two plots do in fact show different analyses. Figure 4E shows the unfolding rate of the PreQ₁ riboswitch in the presence of compound 4, whereas Supp figure 5i show the refolding rate. The unfolding rate is calculated as $\log(1/\text{lifetime}_{\text{folded_state}})$, and the refolding rate is calculated as $\log(1/\text{lifetime}_{\text{unfolded_state}})$. Compound 4 showed a decrease in the unfolding rate and increase in the refolding rate, indicating that compound 4 stabilizes the RNA and therefore it remains longer in the folded state. This is in contrast to the PreQ₁ ligand, where neither the unfolding nor refolding rates are affected, but rather the rate of occurrence of the long folded state which increases with PreQ₁ ligand concentration. This mechanistic distinction is discussed on page 17 of the revised manuscript.

Supplemental. Isn't it more accurate to label the section on Fluorescence Intensity as Fluorescence Anisotropy since this is the basis of the fluorescence changes, which will be more familiar to the reader?

We argue that fluorescence intensity assay is more appropriate for this analysis as we determine the alteration of fluorescence of tagged RNA as a function of small molecule

concentration. In these assays the quenching effect most commonly occurs due to alteration of the local solvent environment near the fluorophore. In contrast, fluorescence anisotropy refers to the changes in fluorescence caused by changes in polarity or orientation of the ligand and accurate measurement requires the use of appropriate dichroic mirrors and plane polarized light (which were not used here). Since we do not make use of any polarized light source or mirrors we prefer to use the fluorescence intensity assay nomenclature.

Supplemental. For the structure determination, the same crystal form appears to have been described previously, which was used for MR. Please state whether the test set was preserved from this previous structure to avoid overfitting of the new structures as recommended [Kleywegt & Brünger (1996) Structure 4, 897]. The authors used TLS parameters in their previous search model. Was TLS refinement conducted here? If so, please state how the TLS groups were chosen. For consideration, the report of Rfree values in the Supplemental text is incomplete without providing the corresponding Rwork values, which should be within 6% to indicate that overfitting did not occur. For example, a reported Rfree of 0.193 with an Rwork of 0.130 would be problematic.

We did not preserve test set reflections from the previous structure. However, alternatively, we performed high temperature simulated annealing to uncouple the working and free R values in the refinement process. Therefore, we are convinced that the current co-crystal structures are refined adequately. To explain this matter clearly, we have revised “X-ray co-crystal structure determination and refinement” section of supplementary information.

As suggested by the Reviewer #3, we also conducted TLS refinement in this study. The TLS groups were identified automatically by the implemented tool in phenix.refine. We have added this matter in the revised version.

Finally, we have revised the Supplemental text to include an Rwork in addition to an Rfree for each co-crystal structure.

Supplementary Fig 1. Many of the curves do not show binding saturation, making it difficult to fit the K_D . This resulted in large errors. The reviewer recommends adding additional concentration points at 1 mM ligand levels. Alternatively, the authors should qualify for the reader that these values are only estimates of affinity due to the lack of saturation. This is acceptable only because these compounds are not the main focus of the manuscript. The figure should also use consistent significant figures.

Thank you for this comment. We agree that it would be better to see more saturation and that apparent K_D is more appropriate to use here. Although we tried these experiments, higher concentrations for many compounds are not possible due to solubility and

aggregation issues, and increased DMSO levels led to RNA denaturation. For **5**, we did reach the concentration of 1mM. We therefore mention in the text that these are apparent K_D of the compounds, as suggested. The revised Legend to Figure 1 now states “Chemical structures of PreQ₁ riboswitch aptamer ligands and their **apparent** binding affinities to a *Bsu*-PreQ₁ aptamer”, and line 124 now states “Data were fitted using a one-site total binding model to measure an **apparent** equilibrium dissociation constant...”.

Revised figure was remade with uniformity in the text.

Supplementary Fig 3. Why is there a doublet for the read-through product? This should be noted in the approach.

Thank you for noting this. This doublet is the result of an occasional run-off product by the *E. coli* RNA polymerase, which we periodically observe in these assays. As the reviewer requested, we have noted this point and the fact that the presence of the doublet does not impact assay quantitation in the revised legend to Figure S4.

Supplementary figure 4. What does “delta” mean? Is it the reactivity of the apo state minus the bound state or vice versa? For clarity please add the riboswitch sequence and secondary structure from the folded state above the delta shape reactivity profiles. This will allow the reader to follow the arguments in the text about regions that change. A particularly difficult aspect about the interpretation of this result is that the reader does not know what regions of the riboswitches are modulating because they have not been provided. To be effective, the reader should have the corresponding secondary structures and their corresponding structural elements must be labeled (e.g., P1, P2, etc, L1, L2 etc). Also, the corresponding expression platforms must be labeled. Please state the significance of the green versus purple highlighting.

Thank you for the comment and we regret not clarifying this earlier. Delta means the SHAPE reactivity of the bound state vs the apo (unbound) state. We have made changes in the figure to clarify this. We have mentioned the regions that are significantly changed in the main text as suggested. We have also mentioned the significance of the two highlights in the figure legend.

Supplementary table 4. The reported I/s(I) is different from the PDB validation reports. Table S4 reports compound 8 as 1.44 but the PDB states 1.43; Table S4 reports compound 4 as 1.57 but the PDB states 1.23. Please correct this error. Also, the Greek letter sigma has been replaced by S from the Latin alphabet. Please use Σ in the footnotes to indicate summation. Also, the PDB Validation Reports provided are not the versions required for manuscript review, meaning the coordinates have not been validated by the PDB. Please provide valid reports.

With this revision, we have submitted the latest version of the PDB validation reports. Reviewer #3 appropriately pointed out the discrepancies of the crystallographic statistics between those in Table S1 and the PDB validation reports, such as $I/\sigma(I)$. We appreciate the reviewer's comment. However, the values in Supplementary Table 4 are derived from the programs we used for data processing. Therefore, we believe that these values from the log files of data processing by the programs KAMO and XDS are correct. Thus, we did not revise these values this time. As pointed out by this reviewer, we have revised “s and S” to use the Greek letter sigma “ σ and Σ ”, respectively, in this revision.

Reference 51 is duplicated from 23. Please check the other references for duplications.

Thank you for noting this. We have checked to ensure duplicate references are not present in the revised manuscript.

REVIEWERS' COMMENTS

Reviewer #1 (Remarks to the Author):

In this revised manuscript, the authors have thoroughly addressed the issues I raised in my previous review. This is an outstanding contribution to the literature, and I fully support publication in Nature Communications.

Reviewer #2 (Remarks to the Author):

The authors have satisfactorily addressed all my concerns, and I support publication in its current form.

Reviewer #3 (Remarks to the Author):

Comments on the Revised Manuscript

Rev. #3

The authors have been very responsive to the requested revisions. The reviewer is satisfied with most responses, but the revisions have raised important points that should be addressed. Overall, this is a nice piece of work that will be of broad interest to the community.

Comments

1). Rev. #3 wrote: Supplementary Fig 1. Many of the curves do not show binding saturation, making it difficult to fit the KD. This resulted in large errors. The reviewer recommends adding additional concentration points at 1 mM ligand levels. Alternatively, the authors should qualify for the reader that these values are only estimates of affinity due to the lack of saturation. This is acceptable only because these compounds are not the main focus of the manuscript. The figure should also use consistent significant figures.

The authors responded: Thank you for this comment. We agree that it would be better to see more saturation and that apparent KD is more appropriate to use here. Although we tried these experiments, higher concentrations for many compounds are not possible due to solubility and aggregation issues, and increased DMSO levels led to RNA denaturation. For 5, we did reach the concentration of 1mM. We therefore mention in the text that these are apparent KD of the compounds, as suggested. The revised Legend to Figure 1 now states "Chemical structures of PreQ1 riboswitch aptamer ligands and their apparent binding affinities to a Bsu-PreQ1 aptamer", and line 124 now states "Data were fitted using a one-site total binding model to measure an apparent equilibrium dissociation constant...".

Revised figure was remade with uniformity in the text.

Rev #3 replied: Use of the word "apparent" is not the same as "estimated" or "approximate". Every reported Kd is an "apparent Kd" unless the exact chemical pathway for the binding event has been rigorously established. In the current manuscript, the reviewer insists that some Kd values can only be considered "approximate" or "estimates" if the curves do not saturate. This is not the same as calling them "apparent" since all binding constants should be called apparent here. Accordingly, please state that certain binding constants in Fig. 1 did not reach saturation and must be considered approximate

or estimates of binding. This approach is more rigorous and transparent because it informs the reader about the values that can truly be trusted versus those that are somewhat qualitative.

2). Rev. #3 wrote: Reference 51 is duplicated from 23. Please check the other references for duplications.

The authors responded: Thank you for noting this. We have checked to ensure duplicate references are not present in the revised manuscript.

Rev. #3 wrote: Reference 23 is duplicated as ref. 55. Please correct the duplication.

3). New comment. Please add larger fonts to the ordinate and abscissa in new figure S7. These are too small to read in the revised manuscript.

4). New comment. In Figure 4, please clarify in the legend that each secondary structure diagram on two left panels represents an experiment conducted in the presence of preQ1 (left) and compound 4 (right). Please state how many times each SHAPE-Map experiment was replicated (as a technical or biological replicate).

5). New comment. For figure 6, please state that the image are representative of biological replicates and note the number of replicates. In general, the authors should make certain that the number of replicates is reported for various experiments, which is usually stated in the legends or the methods.

6). New comment. In figure 6, the authors report an "EV" control. If the vector is empty there would be no riboswitch or GFP, so cells would not be green. Please clarify that this is no-riboswitch control".

7). In general, there are a number of typos in the SI. This file will not be copy edited, so the authors should be sure to carefully proof read it.

8). Rev. #2 wrote: Line 148-152: just a comment: it is always surprising that the replacement of nucleosides by spacer riboses (without bases) for the preQ1 system leads to apparently very reliable results and important insights into ligand binding in pockets. (I also refer to the earlier publication on preQ1 binders by the same authors)

The authors responded: Thank you for noticing this – we also find it an intriguing observation. Since Reviewer #1 also noted this, we felt it was noteworthy to add context to the revised manuscript as well. As described in the response to Reviewer #1: We used a triple-abasic mutant in the crystallography because as reported previously (Connelly et al., Nat. Commun. 2019), when the wild-type Tte-PreQ1 riboswitch was used, we only obtained the structures indistinguishable from the ligand-free form of the RNA, in which the nucleobase of A14 occupies the PreQ1-binding site. This result may be due to the lower affinity for the compound compared to that of the cognate ligand PreQ1. Since the KD and T50 values of compounds 4 and 8 were larger than those for PreQ1, we hypothesized that like the previous work, the crystallography would yield structures indistinguishable from the ligand-free form. Therefore, we used a triple-abasic mutant to obtain the complex structures. However, we believe that the co-crystal structures of the triple-abasic mutant with the compounds exhibit biologically relevant forms based on the present results of comprehensive biochemical analyses and homology to other reported structures.

Rev #3 commented: a potential reason why abasic sites were necessary is because of the high sulfate concentrations needed for crystallization. A more recent analysis of the Tte riboswitch (Schroeder et al 2020) showed that crystals could be prepared from low ionic strength, which allowed resolution of all

nucleotides (ribose + bases) in the L2 loop – i.e., bases deleted in the current study. Because the methylamine group of preQ1 likely carries a positive charge at neutral pH, the sulfate ions interfere with its interaction with the backbone; a sulfate ion was observed to localize to the methylamine moiety in the crystal structures (PDB entry 3q50). The high salt also favors duplex formation, which would be prohibitive to base unstacking in L2, which is needed for compound 4 & 8 binding. It would be interesting to see if the low salt conditions reported by Schroeder et al. could be used to co-crystallize compound 4 or 8 without the use of abasic nucleotides. Of course, this is far beyond the scope of the current investigation. (No response is needed).

Reviewer #1 (Remarks to the Author):

In this revised manuscript, the authors have thoroughly addressed the issues I raised in my previous review. This is an outstanding contribution to the literature, and I fully support publication in Nature Communications.

Thank you again for your helpful and constructive review of our work as well as the kind words.

Reviewer #2 (Remarks to the Author):

The authors have satisfactorily addressed all my concerns, and I support publication in its current form.

Thank you to the reviewer for their extremely helpful consideration of our work.

Reviewer #3 (Remarks to the Author):

Comments on the Revised Manuscript

Rev. #3

The authors have been very responsive to the requested revisions. The reviewer is satisfied with most responses, but the revisions have raised important points that should be addressed. Overall, this is a nice piece of work that will be of broad interest to the community.

Thank you to the reviewer for their help and comments. We have addressed each point below.

Comments

1). Rev. #3 wrote: Supplementary Fig 1. Many of the curves do not show binding saturation, making it difficult to fit the KD. This resulted in large errors. The reviewer recommends adding additional concentration points at 1 mM ligand levels. Alternatively, the authors should qualify for the reader that these values are only estimates of affinity due to the lack of saturation. This is acceptable only because these compounds are not the main focus of the manuscript. The figure should also use consistent significant figures.

The authors responded: Thank you for this comment. We agree that it would be better to see more saturation and that apparent KD is more appropriate to use here. Although we tried these experiments, higher concentrations for many compounds are not possible due to solubility and aggregation issues, and increased DMSO levels led to RNA denaturation. For 5, we did reach the concentration of 1mM. We therefore mention in the text that these

are apparent K_D of the compounds, as suggested. The revised Legend to Figure 1 now states “Chemical structures of PreQ1 riboswitch aptamer ligands and their apparent binding affinities to a Bsu-PreQ1 aptamer”, and line 124 now states “Data were fitted using a one-site total binding model to measure an apparent equilibrium dissociation constant...”.

Revised figure was remade with uniformity in the text.

Rev #3 replied: Use of the word “apparent” is not the same as “estimated” or “approximate”. Every reported K_d is an “apparent K_d ” unless the exact chemical pathway for the binding event has been rigorously established. In the current manuscript, the reviewer insists that some K_d values can only be considered “approximate” or “estimates” if the curves do not saturate. This is not the same as calling them “apparent” since all binding constants should be called apparent here. Accordingly, please state that certain binding constants in Fig. 1 did not reach saturation and must be considered approximate or estimates of binding. This approach is more rigorous and transparent because it informs the reader about the values that can truly be trusted versus those that are somewhat qualitative.

We agree with this subtlety and have include the word approximate as the reviewer notes in the revised manuscript.

2). Rev. #3 wrote: Reference 51 is duplicated from 23. Please check the other references for duplications.

The authors responded: Thank you for noting this. We have checked to ensure duplicate references are not present in the revised manuscript.

Rev. #3 wrote: Reference 23 is duplicated as ref. 55. Please correct the duplication.

The authors responded: Thank you for noting this. We have checked to ensure duplicate references are not present in the revised manuscript.

3). New comment. Please add larger fonts to the ordinate and abscissa in new figure S7. These are too small to read in the revised manuscript.

The revised supplementary information has a larger format vector graphic image that clearly labels axes so that the reader can easily see.

4). New comment. In Figure 4, please clarify in the legend that each secondary structure diagram on two left panels represents an experiment conducted in the presence of preQ1 (left) and compound 4 (right). Please state how many times each SHAPE-Map experiment was replicated (as a technical or biological replicate).

This information has been added to the new versions of Figure 4 and information has been added to the legend to state how many replicates were used to acquire data.

5). New comment. For figure 6, please state that the image are representative of biological replicates and note the number of replicates. In general, the authors should make certain that the number of replicates is reported for various experiments, which is usually stated in the legends or the methods.

This statement has been included for Figure 6 as well as throughout the revised manuscript.

6). New comment. In figure 6, the authors report an “EV” control. If the vector is empty there would be no riboswitch or GFP, so cells would not be green. Please clarify that this is no-riboswitch control”.

We have clarified that this is a no-riboswitch control rather than an empty vector lacking GFP.

7). In general, there are a number of typos in the SI. This file will not be copy edited, so the authors should be sure to carefully proof read it.

We have corrected the SI and fixed all typos we found.

8). Rev. #2 wrote: Line 148-152: just a comment: it is always surprising that the replacement of nucleosides by spacer riboses (without bases) for the preQ1 system leads to apparently very reliable results and important insights into ligand binding in pockets. (I also refer to the earlier publication on preQ1 binders by the same authors)

The authors responded: Thank you for noticing this – we also find it an intriguing observation. Since Reviewer #1 also noted this, we felt it was noteworthy to add context to the revised manuscript as well. As described in the response to Reviewer #1: We used a triple-abasic mutant in the crystallography because as reported previously (Connelly et al., Nat. Commun. 2019), when the wild-type Tte-PreQ1 riboswitch was used, we only obtained the structures indistinguishable from the ligand-free form of the RNA, in which the nucleobase of A14 occupies the PreQ1-binding site. This result may be due to the lower affinity for the compound compared to that of the cognate ligand PreQ1. Since the KD and T50 values of compounds 4 and 8 were larger than those for PreQ1, we hypothesized that like the previous work, the crystallography would yield structures indistinguishable from the ligand-free form. Therefore, we used a triple-abasic mutant to obtain the complex structures. However, we believe that the co-crystal structures of the triple-abasic mutant with the compounds exhibit biologically relevant forms based on the present results of comprehensive biochemical analyses and homology to other reported structures.

Rev #3 commented: a potential reason why abasic sites were necessary is because of the

high sulfate concentrations needed for crystallization. A more recent analysis of the Tte riboswitch (Schroeder et al 2020) showed that crystals could be prepared from low ionic strength, which allowed resolution of all nucleotides (ribose + bases) in the L2 loop – i.e., bases deleted in the current study. Because the methylamine group of preQ1 likely carries a positive charge at neutral pH, the sulfate ions interfere with its interaction with the backbone; a sulfate ion was observed to localize to the methylamine moiety in the crystal structures (PDB entry 3q50). The high salt also favors duplex formation, which would be prohibitive to base unstacking in L2, which is needed for compound 4 & 8 binding. It would be interesting to see if the low salt conditions reported by Schroeder et al. could be used to co-crystallize compound 4 or 8 without the use of abasic nucleotides. Of course, this is far beyond the scope of the current investigation. (No response is needed).

Thank you for the comment here – we are also aware that there are many difficult and subtle technical challenges associated with acquiring atomic resolution structural information about RNA-ligand complexes. It remains unclear to us why the abasic sites are required, though as the reviewer notes it clearly was a useful approach here. We strongly agree that better understanding of these fundamental aspects of RNA structural biology and biophysics, while outside the scope of this specific study, are badly needed to help advance the field.